# VidGuard-R1: AI-Generated Video Detection and Explanation via Reasoning MLLMs and RL

**Kyoungjun Park[1], Yifan Yang[2]\*, Juheon Yi[2], Muhammad Muaz[1], Shicheng Zheng[2],**
**Yifei Shen[2], Dongqi Han[2], Caihua Shan[2], Lili Qiu[1,2]\***

[1]The University of Texas at Austin    [2]Microsoft Research
{kjpark, lili}@cs.utexas.edu, yifanyang@microsoft.com

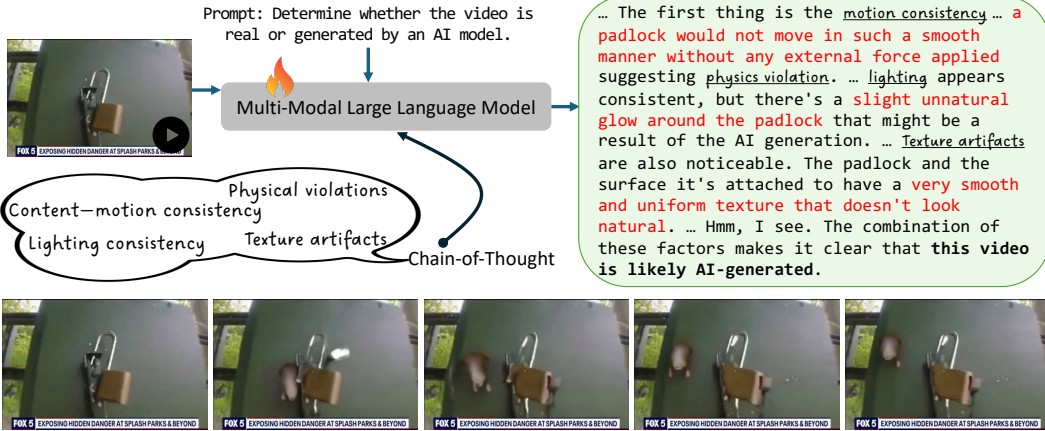

Figure 1: Overall framework of **VidGuard-R1**. We present the first video authenticity detector based on multi-modal large language models (MLLMs), which generates a chain-of-thought reasoning process along with the final answer.

## ABSTRACT

The rapid proliferation of AI-generated video necessitates robust detection tools that offer both high accuracy and human-interpretable explanations. While existing MLLM-based detectors rely on supervised fine-tuning (SFT) or direct preference optimization (DPO), these methods are often bottlenecked by static, pre-labeled datasets that fail to capture the evolving, multi-step physical inconsistencies of modern generative models. To bridge this gap, we introduce **VidGuard-R1**, the first video authenticity detector to utilize group relative policy optimization (GRPO). Moving beyond passive preference matching, **VidGuard-R1** employs a reinforcement learning framework that encourages the model to explore and rank multiple reasoning paths. By introducing specialized reward models for temporal stability and diffusion-aware complexity, we incentivize the model to discover 'physics-grounded' artifacts. Our contributions include: (1) a curated dataset of 140,000 challenging real/fake video pairs; (2) a GRPO-based training paradigm that achieves state-of-the-art zero-shot performance; and (3) a reasoning-first architecture that provides precise, verifiable rationales for its forensic judgments. Project website: https://vidguard-r1.github.io/

## 1 INTRODUCTION

In the past year, we have witnessed unprecedented progress in video generation models, with dramatic improvements in realism and quality. The release of powerful models such as Sora (Brooks et al., 2024), Wan (wan, 2025), and HunyuanVideo (Kong et al., 2024) has made AI-generated videos more accessible to the public, further blurring the line between synthetic videos and real ones. At the same time, these advancements have led to a series of social risks, including the spread of misinformation, violations of privacy rights, damage to personal reputations, and increased susceptibility to scams and fraud.

---

\*Corresponding authors.

Motivated by its practical significance, several pioneering works have been developed to detect AI-generated videos. Early approaches primarily targeted DeepFake-style facial forgeries (Qian et al., 2020; Tan et al., 2024; Gu et al., 2021), which often assumed single-subject, frontal-face scenarios under constrained settings. These assumptions diverge significantly from open-domain, multi-scene videos produced by modern generative models. More recent detectors leverage spatial-temporal consistency (Ma et al., 2024; Bai et al., 2024; Liu et al., 2024); however, such methods are limited in capturing higher-level semantic or causal inconsistencies and can be easily bypassed by post-processing techniques. Other methods are trained on curated fake video detection datasets (Chen et al., 2024a; Ni et al., 2025; Kundu et al., 2025), but these benchmarks often lack coverage of newly emerging models and fail to reflect the full diversity of generative capabilities. A recent benchmark (Chen et al., 2024a) shows that even state-of-the-art detectors still struggle to reliably identify videos from advanced models like Sora. Furthermore, these detectors typically offer only binary decisions without accompanying explanations, which raises concerns for transparency, especially when detection outcomes affect content moderation or legal accountability. Users are also more likely to trust detection systems that provide interpretable reasoning.

Recent advances in multi-modal large language models (MLLMs) have significantly enhanced video understanding, enabling detailed explanations of model decisions (Bai et al., 2023; Zhang et al., 2024b). This makes them promising candidates for detecting and explaining AI-generated videos. However, directly applying existing MLLMs, including advanced models like GPT-4o, yields subpar performance on current benchmarks, underscoring the need for supervised fine-tuning (SFT). As an initial step, we applied SFT to the Qwen2.5-VL-7B model (Bai et al., 2025). While the model achieved strong overall performance, it remained limited in its ability to explain why a video is fake, revealing shortcomings in its reasoning capability.

To address this, we adopt reinforcement learning (RL), which has shown promise in enhancing LLM reasoning (Guo et al., 2025). Notably, Video-R1 (Feng et al., 2025) outperforms commercial models on video reasoning tasks. RL enables MLLMs to develop self-improving reasoning via outcome-based rewards. We hypothesize that RL fine-tuning can help models detect subtle temporal and generative artifacts. Key to this is designing effective reward models. Simple binary rewards (e.g., 1 for real, 0 for fake) are insufficient. Instead, we propose two strategies: (1) injecting temporal artifacts into both real and fake videos to encourage temporal reasoning, and (2) assigning higher rewards to videos generated with more diffusion steps, which are harder to detect. Incorporating these into a group relative policy optimization (GRPO) framework yields over 86% accuracy on our dataset and 95% on two benchmarks.

- We introduce **VidGuard-R1**, the first video authenticity detector that fine-tunes the MLLM using GRPO. The model leverages the pretrained knowledge of MLLMs for accurate classification and employs reinforcement learning for effective exploration. To further enhance performance, we design two specialized reward models that target temporal artifacts and generation complexity based on diffusion steps.

- We construct a challenging dataset of 140k real/fake video pairs for AI-generated video detection. By employing state-of-the-art generation models and carefully controlling the process, we ensure that distinguishing real from fake is non-trivial.

- **VidGuard-R1** achieves state-of-the-art zero-shot performance on existing benchmarks, with accuracy exceeding 95%. Case studies further highlight its ability to produce accurate and interpretable explanations.

## 2 RELATED WORKS

### 2.1 AI-GENERATED VIDEO DETECTION METHOD

Recent research on AI-generated video detection has largely focused on deepfake videos with synthetic faces (Pei et al., 2024), using spatial–temporal consistency, frequency artifacts, or data-driven approaches. These methods often struggle to generalize beyond face-centric content to more diverse, real-world videos. Recently, general video detection methods have emerged: AIGDet (Bai et al., 2024) captures spatial–temporal anomalies, DeCoF (Ma et al., 2024) exploits frame consistency, and diffusion-based representations track temporal dynamics (Liu et al., 2024). Other works identify appearance, motion, and geometry as key factors for classifier training (Chang et al., 2024).

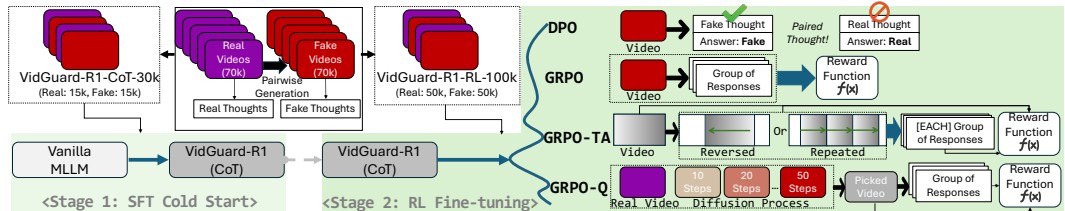

Figure 2: The overall training framework of **VidGuard-R1**, consisting of two stages: (1) supervised fine-tuning (SFT) for chain-of-thought (CoT) initialization, and (2) reinforcement learning-based fine-tuning to enable deeper reasoning.

Recent efforts in forgery detection leverage multimodal LLMs, including FakeShield (Xu et al., 2024), which employs supervised fine-tuning, and SafeWatch (Chen et al., 2024b), which applies direct preference optimization (DPO) for video guardrails. However, DPO relies on static preference pairs that struggle to capture subtle temporal inconsistencies in evolving generative models. We introduce the first application of group relative policy optimization (GRPO) to AI-generated video detection. By enabling iterative exploration and ranking of multiple reasoning paths, our method promotes a deeper physics-aware understanding of video consistency, leading to improved generalization across frontier generative models and diverse benchmarks.

## 2.2 AI-GENERATED VIDEO DETECTION DATASET

Given the recency of this research area, only a limited number of benchmarks have been introduced. The generated video dataset (GVD) (Bai et al., 2024) (11k samples) and GenVideo (Chen et al., 2024a) (with millions of samples) consider settings where both training and test videos are generated by the same series of models. However, these benchmarks lack prompt/image–video pairs, semantic labels, or cross-source settings. GVF (2.8k samples) contains prompts/images–video pairs and semantic labels, but does not provide cross-source settings. GenVidBench (Ni et al., 2025) consists of 100k videos and incorporates cross-source settings, but the video generation models used are less advanced, such as CogVideo and SVD.

Moreover, existing datasets often contain shortcuts in resolution, frame rate, bitrate, or data-source imbalance, enabling models to exploit superficial cues rather than learn intrinsic visual realism. To address these limitations, we construct a curated dataset of 140,000 real–fake videos generated with state-of-the-art video generation models: HunyuanVideo (Kong et al., 2024) and CogVideoX (Yang et al., 2024). Our dataset explicitly standardizes bitrate, resolution, frame rate, and content distribution, resulting in a shortcut-free benchmark that encourages models to rely on semantic and temporal realism rather than on metadata artifacts.

## 3 METHODOLOGY

Figure 2 illustrates the **VidGuard-R1** framework, which consists of two stages. We first apply supervised fine-tuning (SFT) to the multimodal large language model (MLLM), followed by direct preference optimization (DPO) and group relative policy optimization (GRPO) based on the collected datasets. We further develop two GRPO variants by introducing temporal artifacts and leveraging videos generated with varying diffusion steps.

### 3.1 DATA COLLECTION

#### 3.1.1 DATA CONSTRUCTION FOR VIDEO REALISM DISCRIMINATION

High-quality training data is essential for video reasoning in MLLMs. However, many existing benchmarks for real vs. generated video classification, such as GenVideo (Chen et al., 2024a) and GenVidBench (Ni et al., 2025) exhibit uncontrolled discrepancies in basic metadata—e.g., real videos are often longer than 10 seconds, whereas generated videos are typically under 4 seconds in GenVideo. Moreover, they reveal clear modality-level gaps in motion dynamics and content contrasts between real and generated videos. These differences introduce unintended shortcuts, enabling models to rely on superficial cues like duration or resolution rather than actual visual realism. As a result, **VidGuard-R1** attains over 96% accuracy on both GenVideo and GenVidBench by exploiting such artifacts. To mitigate this reward hacking behavior, we construct a curated dataset with standardized video properties, encouraging models to focus on intrinsic visual content.

We collect real videos from the InternVid (Wang et al., 2024) and ActivityNet (Caba Heilbron et al., 2015) datasets and generate their corresponding fake counterparts using HunyuanVideo (Kong et al., 2024) and CogVideoX (Yang et al., 2024). We specifically choose these two models because they support conditioning on both the first-frame image and a text description—an essential requirement for generating videos that are contextually aligned with their real counterparts. To achieve such alignment, we provide the generation models with the first frame of each real video along with a textual caption describing its content. For ActivityNet, which lacks native captions, we extract concise descriptions using Qwen2.5-VL 72B. This pairing strategy mitigates content-based biases and forces the model to reason over subtle visual details.

### 3.1.2 COLLECTING CHAIN OF THOUGHT (CoT) ANNOTATION

Eliciting deliberate, step-by-step reasoning in MLLMs requires high-quality CoT supervision. To this end, we leverage Qwen-2.5-VL (72B) to extract salient visual cues from each video and guide the model toward a deeper understanding. Specifically, we query the model with critical factors known to distinguish real from generated content—motion consistency, lighting consistency, texture artifacts, and physical plausibility violations. These targeted prompts encourage detailed reasoning grounded in visual evidence.

However, current MLLMs lack the capacity to reliably distinguish real from fake videos on their own. To compensate, we provide ground-truth labels during prompt construction and instruct the model to generate CoT rationales conditioned on the given label. While these rationales do not reflect genuine discrimination ability, they capture rich contextual cues—such as object interactions, background details, and lighting inconsistencies—that are highly informative. These CoT annotations serve as useful clues for subsequent reinforcement learning fine-tuning. For prompt templates used in CoT generation, please refer to our supplementary materials.

### 3.2 SUPERVISED AND RL FINE-TUNING

We begin with SFT, where the model is trained to mimic the ground-truth reasoning process. Given a video $x$ and its annotation $y$ from the collected dataset, the model parameters $\theta$ are optimized by minimizing the negative log-likelihood $\mathcal{L}_{\text{SFT}}(\theta) = -\sum_{t=1}^{T} \log p_\theta(y_t \mid y_{<t}, x)$. To align the model outputs with human preferences, we apply DPO, which updates the model based on pairwise preference data without explicit reward modeling. Given a preferred response $y_w$ and a less-preferred response $y_l$ for the same video $x$, the DPO loss encourages the model to prefer $y_w$ over $y_l$ compared to a reference model $p_{\text{ref}}$:

$$\mathcal{L}_{\text{DPO}}(\theta) = -\mathbb{E}_{(x,y_w,y_l)\sim D}\left[\log \sigma\left(\beta \log \frac{p_\theta(y_w|x)}{p_{\text{ref}}(y_w|x)} - \beta \log \frac{p_\theta(y_l|x)}{p_{\text{ref}}(y_l|x)}\right)\right]$$

where $\sigma(\cdot)$ is the sigmoid function and $\beta$ controls the preference strength. This method allows fine-tuning using preference comparisons without requiring scalar rewards.

Finally, we adopt GRPO from DeepSeek R1 (Guo et al., 2025), which generalizes RLHF to group-level comparisons. Given a query video $x$ and a group of generated outputs $\{o_i\}_{i=1}^{G}$, the model is trained to assign higher probabilities to outputs with higher rewards. The GRPO objective is:

$$\mathcal{L}_{\text{GRPO}}(\theta) = -\mathbb{E}_{(x,o_{1:G})\sim D}\left[\frac{1}{G}\sum_{i=1}^{G}\min\left(\frac{p_\theta(o_i|x)}{p_{\text{ref}}(o_i|x)}A_i, \text{clip}\left(\frac{p_\theta(o_i|x)}{p_{\text{ref}}(o_i|x)}, 1-\epsilon, 1+\epsilon\right)A_i\right) - \beta D_{\text{KL}}\left(p_\theta \,\|\, p_{\text{ref}}\right)\right]$$

where $\epsilon$ is a clipping threshold and $\beta$ regularizes the policy to stay close to the reference model. The advantage term $A_i$ normalizes the reward $r_i$ for output $o_i$ within the group, computed as $A_i = \frac{r_i - \mu_x}{\sigma_x}$, where $\mu_x$ and $\sigma_x$ are the mean and standard deviation of $\{r_i\}_{i=1}^{G}$. GRPO thus enables learning from relative ranking among multiple responses, capturing nuanced distinctions in quality across outputs.

### 3.3 VIDGUARD-R1

#### 3.3.1 OVERVIEW

Figure 2 illustrates the training pipeline of **VidGuard-R1**. Following the data collection procedure, we construct two datasets of different scales: `VidGuard-R1-CoT-30k` and `VidGuard-R1-RL-100k`. We adopt Qwen2.5-VL-7B as the base MLLM and train it using our proposed fine-tuning framework.

The first stage is supervised fine-tuning initialization using the `VidGuard-R1-CoT-30k` dataset, which contains videos paired with chain-of-thought (CoT) annotations. This stage establishes foundational reasoning ability and equips the model with basic cross-modal alignment and visual understanding. The resulting model is referred to as **VidGuard-R1 (CoT)**.

In the second stage, we apply two reinforcement learning methods—DPO and GRPO—to further refine the model on a larger and more diverse dataset, `VidGuard-R1-RL-100k`. DPO aligns the model with high-quality preference signals via pairwise comparisons, requiring the construction of preference pairs. Specifically, since our dataset includes pairwise real and fake videos, each sample is annotated with CoT rationales for both perspectives. For DPO training, we construct preference pairs by swapping these CoTs. For a real video, the CoT supporting its authenticity with the answer "real" serves as the positive annotation, while the CoT from the paired fake video with the answer "fake" is used as the negative annotation. In contrast, GRPO encourages consistent performance across grouped outputs by leveraging structural regularization. As it does not rely on preference annotations, video labels are directly used as reward signals. The resulting models are denoted as **VidGuard-R1 (DPO)** and **VidGuard-R1 (GRPO)**.

We introduce two variants, GRPO-TA and GRPO-Q, to further enhance detection performance. These methods extend the original GRPO framework by adjusting reward values according to the difficulty of detecting fake videos. Detailed descriptions are provided in the following sections.

### 3.3.2 GRPO WITH TEMPORAL ARTIFACTS (GRPO-TA)

While standard GRPO performs well in video discrimination by leveraging local visual cues—such as pixel distortions and lighting inconsistencies—it often overlooks temporal inconsistencies, which are crucial for detecting generated videos. To address this limitation, we introduce **GRPO with temporal artifacts (GRPO-TA)**, a variant that explicitly promotes temporal reasoning through a contrastive reward adjustment.

We apply two common temporal artifacts: (1) repeating a specific video segment and (2) reversing the frame sequence within a segment. These manipulations are applied probabilistically, with the manipulated region selected based on a Gaussian distribution over the video timeline.

Specifically, for each input query, we generate two sets of model outputs: $\{o_i\}_{i=1}^{G}$ for the original video, and $\{\tilde{o}_i\}_{i=1}^{G'}$ for the corresponding manipulated video with temporal artifacts. These videos should be classified as fake videos. In GRPO-TA, we assign additional rewards when the model correctly classifies temporally manipulated videos as fake. Consider two numbers, $\alpha_1 > \alpha_2$. Detecting temporal artifacts in videos manipulated from real content tends to be more challenging than identifying those derived from fake videos. This is because real videos typically exhibit coherent and natural motion, so temporal manipulations such as frame shuffling or repetition can be subtle and difficult to detect. In contrast, generated videos often contain artifacts like unstable motion or low temporal consistency, which make further manipulations more visually salient. To reflect this asymmetry in difficulty, we assign the model a higher reward $\alpha_1$ when the original video $o_i$ is real, and a moderate reward $\alpha_2$ when the original video is fake. This is defined as:

$$w_i = \begin{cases} \alpha_1, & \text{if } \tilde{o}_i = \textit{fake} \text{ and } y_i = \textit{real} \\ \alpha_2, & \text{if } \tilde{o}_i = \textit{fake} \text{ and } y_i = \textit{fake} \\ 0, & \text{otherwise} \end{cases} \tag{1}$$

where $y_i$ denotes the label of the $i$-th video. In the experiments, we set the hyperparameters to $\alpha_1 = 0.5$ and $\alpha_2 = 0.3$. This additional reward, $w_i$, is designed to be applied conditionally. Specifically, for a given sample, we only add $w_i$ to the original GRPO reward if two conditions are met: the model's prediction on the original video ($O_i$) must be correct, and the overall accuracy on the group of manipulated videos ($\tilde{p}$) must exceed a predefined threshold $\mu$. This ensures that we only reward the model for temporal reasoning when it already has a solid baseline performance. The final reward of GRPO-TA is given by

$$r_i^{\text{GRPO-TA}} = \begin{cases} r_i^{\text{GRPO}} + w_i, & \text{if } o_i \text{ is correct and } \tilde{p} > \mu \\ r_i^{\text{GRPO}}, & \text{otherwise} \end{cases} \tag{2}$$

where $r_i$ denotes the original GRPO reward, set to 1 if the model prediction on the original video is correct, and 0 otherwise. The additional reward $w_i$ is applied only when both the original prediction is correct and the group of responses for the temporally manipulated videos achieves higher accuracy. In the experiments, we set $\mu = 0.8$.

### 3.3.3 GRPO WITH QUALITY EVOLUTIONARY VIDEOS (GRPO-Q)

Our motivation is to extend the model's capability to detect videos based on quality. Given the subjective nature of quality assessment, we avoid relying on large-scale human annotations. Instead, we leverage diffusion-based video generation by systematically varying the number of reverse diffusion steps to produce videos with distinct quality levels.

As in GRPO-TA, $o_i \in \mathcal{Y}$ and $y_i \in \mathcal{Y}$ denote the model output and ground-truth label, with $\mathcal{Y} = \{real\} \cup \{fake\text{-}s\}$, where $s$ is the diffusion step. A reward is given for an exact match, and no reward is assigned if the real/fake classification is incorrect. In GRPO-Q, if the model correctly classifies a fake video but selects an incorrect diffusion step, we assign a partial reward based on the distance between the predicted and ground-truth diffusion steps. The GRPO-Q reward is defined as follows:

$$r_i^{\text{GRPO-Q}} = \begin{cases} 0, & \text{if } \big(o_i = real \text{ and } y_i \neq real\big) \text{or} \big(o_i \neq real \text{ and } y_i = real\big) \\ \delta, & \text{if } o_i = y_i \\ |g\,(o_i, y_i)|\,, & \text{if } o_i, y_i \in \mathcal{Y} \setminus \{real\}. \end{cases} \quad (3)$$

The first scenario occurs when the model fails to correctly classify the video as real or fake. The second scenario, where $\delta = 1$, represents an exact match in prediction, including the diffusion progression. In the third case, the function $g(\cdot, \cdot)$ maps the step distance to a scalar reward, enabling fine-grained credit assignment based on the similarity in quality. Specifically, we define a progress value $s()$ in the range $[0, 1]$ to indicate the fraction of diffusion steps used, where 0 denotes zero steps, and 1 denotes full completion of the steps. The ground-truth value is $s(y_i)$, and the model will estimate a progress value. We define the reward function as $g(o_i, y_i) = \delta \cdot (1 - |s(o_i) - s(y_i)|)$.

This reward formulation enables the model to move beyond binary discrimination and perform fine-grained analysis of video quality. By learning to associate subtle differences in generation steps with quality variations, the model develops a deeper understanding of the diffusion process and its impact on perceptual realism. As a result, it can not only detect whether a video is fake, but also infer and estimate the degree of quality degradation in generated videos. This facilitates more interpretable and controllable evaluation of generated content quality.

## 4 EXPERIMENTS

### 4.1 IMPLEMENTATION DETAILS

#### 4.1.1 DATASET

Our dataset contains 140k videos, balanced between 70k real and 70k generated samples, organized into contextual pairs. The real set comprises 55k videos from InternVid and 15k from ActivityNet, while the generated set includes 50k samples synthesized by HunyuanVideo-I2V (Kong et al., 2024) and 20k by CogVideoX-5B (Yang et al., 2024). We allocate 130k samples for training and 10k for testing, with the latter evenly split between real and generated videos. Within the training data, 30k samples are reserved for chain-of-thought (CoT) learning, denoted as `VidGuard-R1-CoT-30k`, and the remaining 100k are used for reinforcement learning fine-tuning, denoted as `VidGuard-R1-RL-100k`.

Since the state-of-the-art generative models still produce relatively short videos ($\sim$129 frames) at modest resolutions, we standardize all real videos to match generated ones by enforcing 49 frames, 8 FPS, 720×480 resolution, and YUV420p format.

For GRPO-Q fine-tuning, we augment the training set with intermediate generations sampled from diffusion steps 10 to 50. These are labeled with approximate quality levels (20%, 40%, 60%, 80%, and 95%). Specifically, we use 12k real videos, each paired with five generated variants at different diffusion steps, resulting in 72k samples per generation model.

#### 4.1.2 EVALUATION PROTOCOL

We evaluate three datasets—ours, GenVidBench (Ni et al., 2025), and GenVideo (Chen et al., 2024a)—using the metrics and baselines defined by their respective benchmarks. For ours and GenVidBench, we report **mean Top-1 accuracy**, the average correctness over all predictions. For GenVideo, we follow the original protocol and report **recall** and **F1 score**. All evaluations adhere to the official settings of each benchmark to ensure fair comparison.

Table 1: Comparison of models on our dataset, reported as mean Top-1 accuracy (%). TF denotes transformer.

| Method | Type | CogVideoX | HunyuanVideo |
|---|---|---|---|
| SlowFast | CNN | 77.87 | 77.03 |
| I3D | CNN | 64.78 | 62.13 |
| TRN | CNN | 68.73 | 69.87 |
| UniFormer V2 | TF | 73.95 | 71.92 |
| TimeSformer | TF | 78.53 | 74.55 |
| VideoSwin | TF | 76.81 | 79.71 |
| MViT V2 | TF | 58.38 | 53.91 |
| Qwen2.5-VL-7B | MLLM | 50.95 | 52.83 |
| Qwen2.5-VL-72B | MLLM | 54.17 | 55.82 |
| GPT-4.1 mini | MLLM | 54.95 | 55.31 |
| GPT-4o | MLLM | 56.81 | 57.42 |
| VidGuard-R1 (CoT) | MLLM | 66.18 | 63.19 |
| VidGuard-R1 (DPO) | MLLM | 79.13 | 80.88 |
| VidGuard-R1 (GRPO) | MLLM | 81.30 | 81.90 |
| VidGuard-R1 (GRPO-TA) | MLLM | 82.17 | 83.72 |
| VidGuard-R1 (GRPO-Q) | MLLM | **84.32** | **86.17** |

Table 2: Extended GenVidBench results with **VidGuard-R1** and additional MLLMs, reported as mean Top-1 accuracy (%). TF denotes transformer.

| Method | Type | MuseV | SVD | CogVideo | Mora | HD-VG | Mean |
|---|---|---|---|---|---|---|---|
| SlowFast | CNN | 12.25 | 12.68 | 38.34 | 45.93 | 93.63 | 41.66 |
| I3D | CNN | 8.15 | 8.29 | 60.11 | 59.24 | 93.99 | 49.23 |
| TRN | CNN | 38.92 | 26.64 | 91.34 | 93.98 | 93.97 | 71.26 |
| UniFormer V2 | TF | 20.05 | 14.81 | 45.21 | 99.21 | 96.89 | 57.55 |
| TimeSformer | TF | 73.14 | 20.17 | 74.80 | 39.40 | 92.32 | 64.28 |
| VideoSwin | TF | 62.29 | 8.01 | 91.82 | 45.83 | **99.29** | 67.27 |
| MViT V2 | TF | 76.34 | **98.29** | 47.50 | 96.62 | 97.58 | 79.90 |
| Qwen2.5-VL-7B | MLLM | 25.86 | 27.06 | 68.51 | 43.26 | 71.15 | 47.30 |
| GPT-4.1 mini | MLLM | 26.07 | 33.78 | 94.07 | 57.19 | 87.64 | 59.62 |
| VidGuard-R1 (CoT) | MLLM | 36.52 | 16.02 | 99.35 | 76.94 | 99.94 | 66.09 |
| VidGuard-R1 (GRPO, GenVideo-pretrained, Zero-shot) | MLLM | 97.24 | 96.59 | 99.88 | 99.93 | 88.14 | 96.37 |
| VidGuard-R1 (GRPO) | MLLM | **97.38** | 94.98 | **99.90** | **99.99** | 95.46 | **97.53** |

### 4.1.3 TRAINING SETUP

We employ Qwen2.5-VL-7B as the base MLLM and conduct all experiments on four NVIDIA A100 GPUs (80GB). Each video is represented by up to 16 frames, each resized to $28 \times 28$ and mapped to 128 feature channels for encoder input during both training and inference. For GenVideo and GenVidBench, we follow their official evaluation protocols and adopt 8-frame inputs. In GRPO training, we sample 8 responses per input; for GRPO-TA, we additionally sample 4 responses from temporally manipulated variants of the input to enhance robustness against temporal artifacts. Training proceeds in two stages: first, the base model is fine-tuned for one epoch on the CoT dataset, yielding the SFT-CoT MLLM; second, we initialize **VidGuard-R1** with SFT-CoT and perform reinforcement learning for approximately 2,000 steps.

### 4.2 MAIN RESULTS

### 4.2.1 OUR DATASET

We evaluate **VidGuard-R1** on our dataset with several methods, including CNN-based models (SlowFast (Feichtenhofer et al., 2019), I3D (Carreira & Zisserman, 2017), TRN (Zhou et al., 2018)), Transformer-based models (UniFormer V2 Li et al. (2022a), TimeSformer (Bertasius et al., 2021), VideoSwin (Liu et al., 2022), MViT V2 (Li et al., 2022b)), and MLLM-based models (Qwen2.5-VL (Bai et al., 2025) and GPT-4.1 mini (OpenAI, 2025)). For CNN and Transformer models, we use the default training settings provided by the MMAction2 framework (Contributors, 2020).

As shown in Table 1, CNN- and Transformer-based models achieved 53–79% accuracy, with Slow-Fast and TimeSformer among the top performers. In contrast, Qwen2.5-VL-7B and GPT-4.1 mini exhibited near-random performance, highlighting their limited capability in distinguishing fake videos. *VidGuard-R1 (CoT)*, trained via supervised fine-tuning (SFT) on Qwen2.5-VL-7B, substantially improved accuracy from around 51% to over 66%, yet remained less competitive compared to advanced SOTA methods. This result aligns with the intended role of the SFT stage—as a cold start phase to guide the model toward structured *think + answer* responses, emphasizing the extraction of salient visual cues.

In the subsequent RL stage, both DPO and GRPO further improved performance by roughly 2% over the best baseline. Our proposed methods—GRPO-TA and GRPO-Q—achieved additional gains of approximately 2% and 5% over GRPO, respectively, demonstrating the effectiveness of temporal artifact supervision and quality-aware reward modeling in enhancing detection accuracy.

### 4.2.2 GENVIDBENCH BENCHMARK

The GenVidBench dataset comprises approximately 87k training samples and 82k testing samples, with fake videos generated by models such as MuseV (Xia et al., 2024), SVD (Blattmann et al., 2023), CogVideo (Hong et al., 2022), and Mora (Yuan et al., 2024), and real videos sourced from HD-VG (Wang et al., 2023b). We conduct training and evaluation under the cross-source and cross-generator settings as proposed in their benchmark. In addition to the models originally reported in GenVidBench, we evaluate **VidGuard-R1** using the same model families as in our dataset experiments—CNN-based, Transformer-based, and MLLM-based models—including two MLLMs: Qwen2.5-VL and GPT-4.1 mini. `VidGuard-R1 (GRPO, GenVideo-pretrained, Zero-shot)` denotes the zero-shot model pretrained on GenVideo

Table 3: Extended GenVideo results with **VidGuard-R1** and additional MLLMs, evaluated by F1 score and recall (R)

| Method | Detection level | Metric | Sora | Morph Studio | Gen2 | HotShot | Lavie | Show-1 | Moon Valley | Crafter | Model Scope | Wild Scrape | Mean |
|---|---|---|---|---|---|---|---|---|---|---|---|---|---|
| NPR (Tan et al., 2024) | Image | R | 0.91 | 0.99 | 0.99 | 0.24 | 0.89 | 0.57 | 0.97 | 0.99 | 0.94 | 0.87 | 0.84 |
| | | F1 | 0.27 | 0.84 | 0.91 | 0.30 | 0.86 | 0.59 | 0.81 | 0.91 | 0.81 | 0.81 | 0.71 |
| VideoMAE (Tong et al., 2022) | Video | R | 0.67 | 0.96 | 0.98 | 0.96 | 0.77 | 0.80 | 0.97 | 0.96 | 0.96 | 0.68 | 0.87 |
| | | F1 | 0.62 | 0.95 | 0.98 | **0.96** | 0.86 | 0.87 | 0.96 | 0.97 | **0.96** | 0.79 | 0.89 |
| MINTIME-CLIP (Coccomini et al., 2024) | Video | R | 0.89 | 1.00 | 0.98 | 0.26 | 0.96 | 0.98 | 0.99 | 1.00 | 0.84 | 0.82 | 0.87 |
| | | F1 | 0.49 | 0.93 | 0.96 | 0.37 | 0.94 | 0.92 | 0.92 | 0.96 | 0.84 | 0.85 | 0.82 |
| FTCN-CLIP (Zheng et al., 2021) | Video | R | 0.87 | 1.00 | 0.98 | 0.17 | 0.97 | 0.91 | 1.00 | 1.00 | 0.85 | 0.82 | 0.86 |
| | | F1 | 0.78 | 0.98 | 0.98 | 0.29 | 0.98 | 0.94 | 0.98 | **0.99** | 0.90 | 0.89 | 0.87 |
| DeMamba-XCLIP (Chen et al., 2024a) | Video | R | 0.98 | 1.00 | 0.99 | 0.65 | 0.94 | 0.98 | 1.00 | 1.00 | 0.92 | 0.89 | 0.93 |
| | | F1 | 0.64 | 0.96 | 0.97 | 0.75 | **0.95** | 0.95 | 0.95 | 0.97 | 0.92 | **0.91** | 0.90 |
| Qwen2.5-VL-7B (Bai et al., 2025) | MLLM | R | 0.58 | 0.56 | 0.54 | 0.33 | 0.43 | 0.38 | 0.81 | 0.63 | 0.51 | 0.70 | 0.54 |
| | | F1 | 0.74 | 0.72 | 0.70 | 0.49 | 0.60 | 0.55 | 0.90 | 0.77 | 0.68 | 0.82 | 0.70 |
| GPT-4.1 mini (OpenAI, 2025) | MLLM | R | 0.43 | 0.67 | 0.56 | 0.54 | 0.63 | 0.56 | 0.92 | 0.67 | 0.69 | 0.69 | 0.65 |
| | | F1 | 0.60 | 0.80 | 0.72 | 0.70 | 0.77 | 0.72 | 0.96 | 0.80 | 0.82 | 0.82 | 0.72 |
| VidGuard-R1 (CoT) | MLLM | R | 0.92 | 0.89 | 0.91 | 0.90 | 0.98 | 0.79 | 0.99 | 0.85 | 0.89 | 0.87 | 0.90 |
| | | F1 | 0.90 | 0.91 | 0.95 | 0.89 | 0.99 | 0.81 | 0.95 | 0.89 | 0.85 | 0.88 | 0.90 |
| VidGuard-R1 (GRPO, GenVidBench-pretrained, Zero-shot) | MLLM | R | 0.95 | 0.98 | 0.90 | 0.89 | 0.97 | 0.85 | 0.99 | 0.93 | 0.81 | 0.87 | 0.92 |
| | | F1 | 0.93 | 0.93 | 0.96 | 0.91 | 0.99 | 0.82 | 0.95 | 0.89 | 0.85 | 0.88 | 0.91 |
| VidGuard-R1 (GRPO) | MLLM | R | 0.95 | 1.00 | 0.98 | 0.94 | 0.98 | 0.95 | 0.97 | 0.99 | 0.94 | 0.91 | 0.96 |
| | | F1 | **0.97** | **0.99** | **0.99** | 0.91 | **0.99** | 0.89 | **0.99** | **0.99** | 0.95 | 0.90 | **0.96** |

and evaluated on GenVidBench. As shown in Table 2, both the zero-shot model and two fine-tuned variants achieve over 15% higher accuracy compared to prior SOTA methods. Notably, the zero-shot model demonstrates strong generalization, highlighting the effectiveness of pretraining on diverse generative content. Complete detection model results are provided in Appendix B.

### 4.2.3 GENVIDEO BENCHMARK

The GenVideo dataset comprises approximately 2.2M training samples and 20k testing samples, with generated videos sourced from a diverse set of models, including Sora (sor, 2025), MorphStudio (mor, 2025), Gen2 (Esser et al., 2023), HotShot (hot, 2025), Lavie (Wang et al., 2025), Show-1 (Zhang et al., 2024a), MoonValley (moo, 2025), Crafter (Chen et al., 2023), ModelScope (Wang et al., 2023a), and WildScrape (Wei et al., 2024). Following the official evaluation protocol, we benchmark two MLLM baselines and three variants of **VidGuard-R1**. Among these, `VidGuard-R1 (GRPO)` consistently outperforms almost all prior detection methods across videos generated by the various models. As shown in Table 3, it achieves an F1 score improvement of 0.06 compared to DeMamba-XCLIP. Complete detection model results are provided in Appendix B.

### 4.2.4 PERFORMANCE GAP BETWEEN OUR DATASET AND BENCHMARKS

While **VidGuard-R1** achieves approximately 85% accuracy on our dataset, it obtains significantly higher accuracy—exceeding 95%—on the two benchmark datasets. This discrepancy arises from two key differences. First, the benchmarks exhibit clear discrepancies in video metadata—such as resolution, duration, and frame rate—between real and fake videos, which models can exploit as superficial cues. In contrast, we standardize all videos in our dataset by matching resolution, FPS, and format, thereby forcing models to rely on actual visual content. Second, our dataset ensures strong contextual alignment by conditioning generation on the first frame and the corresponding caption of a real video, resulting in more realistic and semantically consistent outputs. In comparison, benchmark datasets often generate fake videos from unrelated prompts and synthetic images, leading to artifacts that make detection easier.

### 4.2.5 ABLATION STUDY

**Explanation quality and accuracy comparison.** Table 4 presents results on the HunyuanVideo (Kong et al., 2024) and CogVideoX (Yang et al., 2024) datasets. We report explanation quality scores, which are rated on a 1–10 scale (with 10 indicating excellent quality and full alignment) by GPT-4.1 mini using the LLM-as-a-judge prompt described in Appendix D. Compared to baseline models such as Qwen2.5-VL-7B and GPT-4.1 mini, our VidGuard-R1 GRPO variants achieve consistent improvements in both classification accuracy and explanation quality.

Table 4: LLM-as-a-judge explanation scores on our dataset

| Method | Expl. Score (HunyuanVideo) | Expl. Score (CogVideoX) |
|---|---|---|
| Qwen2.5-VL-7B | 5.8 | 5.6 |
| GPT-4.1 mini | 5.8 | 5.9 |
| VidGuard-R1 (CoT) | 6.8 | 6.9 |
| VidGuard-R1 (DPO) | 7.2 | 8.1 |
| VidGuard-R1 (GRPO) | 8.1 | 8.0 |
| VidGuard-R1 (GRPO-TA) | 8.1 | 8.4 |
| VidGuard-R1 (GRPO-Q) | **8.3** | **8.5** |

Table 6: Accuracy (%) for **GRPO-Q** with varying number of intermediate diffusion steps

| # of steps (step numbers) | Accuracy (%) |
|---|---|
| 1 (50) | 81.63 |
| 3 (10, 30, 50) | 83.21 |
| **5 (10, 20, 30, 40, 50)** | **85.80** |

Table 7: Cross-dataset evaluation for **VidGuard-R1**

| Test Dataset | Training Source | Accuracy (%) |
|---|---|---|
| Ours | **VidGuard-R1** (Ours only) | 81.65 |
| Ours | **VidGuard-R1** (Ours + GenVideo) | 82.97 |
| GenVideo | **VidGuard-R1** (GenVideo only) | 97.53 |
| GenVideo | **VidGuard-R1** (Ours + GenVideo) | 97.98 |

Table 8: Label–rationale consistency and explanation quality for 20 videos

| Metric | Value |
|---|---|
| Annotators | 5 |
| Label–rationale alignment | 89% |
| Rationale score (0–5) | 3.9 |

Table 9: Human ranking of explanation quality

| Model | Avg. Rank |
|---|---|
| **VidGuard-R1** (GRPO) | 1.67 |
| GPT-4o | 2.08 |
| Qwen2.5-VL-72B | 2.22 |

Table 5: Accuracy (%) for **GRPO-TA** under different reward function parameters $\alpha_1$ and $\alpha_2$

| $\alpha_1$ | $\alpha_2$ | Accuracy (%) |
|---|---|---|
| 0.3 | 0.1 | 81.31 |
| 0.3 | 0.3 | 82.59 |
| **0.5** | **0.3** | **83.57** |
| 0.5 | 0.5 | 83.12 |
| 0.7 | 0.5 | 82.53 |

**GRPO-TA reward ablation.** Table 5 reports an ablation study of GRPO-TA on our dataset by varying the weight parameters $\alpha_1$ and $\alpha_2$, which control the relative importance of different temporal artifact types. The highest classification accuracy of 83.57% is achieved with $\alpha_1 = 0.5$ and $\alpha_2 = 0.3$, while the threshold $\mu$ is fixed at 0.8 across all experiments.

**GRPO-Q reward ablation.** Table 6 presents an ablation study on GRPO-Q conducted on our dataset by varying the number of intermediate diffusion steps included per real video during fine-tuning. Using more steps provides richer supervision of video quality progression, improving detection accuracy. The best accuracy of 85.80% is obtained with five steps, which is the setting used in our main experiments.

**Cross-dataset complementarity.** To assess whether training on a limited generative source induces overfitting, we conduct dataset-mixing experiments using **VidGuard-R1** (GRPO). As shown in Table 7, augmenting our dataset with GenVideo leads to consistent performance gains across both evaluation sets, suggesting that the model benefits from heterogeneous training data and does not rely on artifacts from any single source. These findings indicate that incorporating diverse generative sources enhances overall accuracy, reinforcing that **VidGuard-R1** learns generalizable detection cues rather than dataset-specific patterns.

## 4.3 HUMAN EVALUATION OF EXPLANATION QUALITY

To examine the coherence and interpretability of CoT rationales, we conducted a human evaluation of **VidGuard-R1** (GRPO). Tables 8 and 9 summarize the two complementary studies.

**Consistency and quality.** Five annotators evaluated twenty randomly selected fake videos that the model correctly identified. Annotators judged whether each rationale was consistent with the predicted label and assigned a quality score on a 0–5 scale after watching the corresponding video. As shown in Table 8, annotators reported 89% label–rationale agreement with an average quality score of 3.9. Lower scores ($\leq 2$) occurred primarily when the model emphasized subtle texture-level artifacts that were difficult for human raters to perceive.

**Explanation quality ranking.** We conducted a blind ranking study in which five participants evaluated explanations from three models across twenty videos, consisting of ten real and ten synthetic videos. Table 9 shows that **VidGuard-R1** achieved the strongest human preference, with the lowest average rank (1.67), outperforming GPT-4o and Qwen2.5-VL-72B. Participants consistently judged **VidGuard-R1**'s explanations to be more informative and higher quality. This improvement is likely because, although GPT-4o and Qwen2.5-VL-72B also describe video features well, their explana-

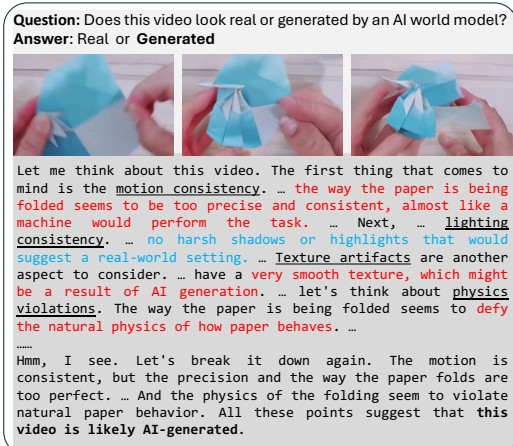

Figure 3: VidGuard-R1 (GRPO): reasoning about an origami folding sequence

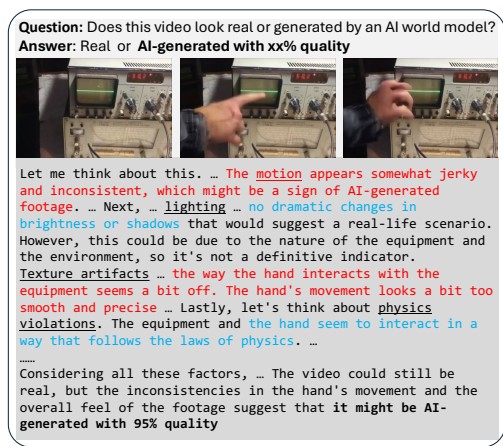

Figure 4: VidGuard-R1 (GRPO-Q): temporal inconsistency observed as unnatural movements

tions are less effective at distinguishing real from synthetic content, whereas **VidGuard-R1** better highlights the discriminative aspects relevant to this task.

**Interpretation and broader implications.** Although the CoT rationales align closely with human judgments, they are not guaranteed to be correct in every case. Hallucinations may arise, and the rationales should therefore be shown with an appropriate disclaimer noting their AI-generated nature. Nevertheless, presenting interpretable cues helps users more effectively assess whether a video may be AI-generated, supporting a practical human-in-the-loop verification process.

### 4.4 CASE STUDIES ON EXPLANATIONS

Figures 3 and 4 illustrate cases where **VidGuard-R1** correctly identifies videos as generated. The model performs multi-faceted reasoning across motion, lighting, texture, and physical plausibility before arriving at a final decision. Notably, it does not rely on a single cue, but instead accumulates evidence across frames, resembling how humans distinguish fake videos. In each figure, pink highlights denote cues suggesting realism, red indicates artifacts indicative of generation, yellow marks intermediate reasoning steps, and underlines represent several key factors.

For instance, in Figure 3, the smooth hand motion initially suggests realism; however, once the origami folds in a physically implausible manner, the model revises its judgment. In Figure 4, although the lighting and shadows are consistent—typically a cue for authenticity—the model recognizes that this is insufficient in a largely static scene with only a stationary machine and a human hand. In particular, even in its final prediction, the model reflects on earlier realistic cues and acknowledges that *the video could still be real*, underscoring its nuanced, human-like reasoning in assessing video quality. Additional case studies are provided in the Appendix E.

## 5 CONCLUSION

We propose **VidGuard-R1**, an MLLM-based discriminator that not only detects AI-generated videos with high accuracy but also provides interpretable reasoning. By leveraging reinforcement learning with reward models targeting temporal artifacts and generation quality, **VidGuard-R1** achieves 85% accuracy on our dataset, 97% on GenVidBench, and 96% on GenVideo, substantially surpassing prior state-of-the-art methods. We expect this work to advance MLLM-based video analysis and foster future research on strengthening MLLMs' reasoning.

### 5.1 LIMITATIONS

Our dataset currently contains fake videos generated by HunyuanVideo and CogVideoX, the main open source models that support large-scale text-image joint conditioning. Most other diffusion models offer only text or image conditioning, making them less suitable for our pairwise data construction. Although this design ensures strong contextual alignment, adding outputs from more generative models would increase diversity and robustness, improving real-world applicability.

ETHICS STATEMENT

This work does not involve personally identifiable information or sensitive user data. All datasets used in our experiments are publicly available and constructed in accordance with their licenses and usage guidelines. VidGuard-R1 is designed to mitigate societal risks associated with AI-generated videos, such as misinformation and reputational harm, by providing interpretable CoT reasoning to assist human verification. To the best of our knowledge, the method does not introduce risks related to fairness, safety, or privacy.

ACKNOWLEDGEMENT

This work was supported in part by the National Science Foundation under Grant CNS-2212297. We are grateful to the anonymous reviewers for their insightful suggestions, which helped improve the quality and clarity of this paper.

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

Table 10: Extended GenVidBench results with **VidGuard-R1** and additional MLLMs, reported as mean Top-1 accuracy (%). TF denotes transformer.

| Method | Type | MuseV | SVD | CogVideo | Mora | HD-VG | Mean |
|---|---|---|---|---|---|---|---|
| SlowFast (Feichtenhofer et al., 2019) | CNN | 12.25 | 12.68 | 38.34 | 45.93 | 93.63 | 41.66 |
| F3Net (Qian et al., 2020) | CNN | 37.43 | 37.27 | 36.46 | 39.59 | 52.76 | 42.52 |
| I3D (Carreira & Zisserman, 2017) | CNN | 8.15 | 8.29 | 60.11 | 59.24 | 93.99 | 49.23 |
| CFV2 (Nguyen et al., 2019) | CNN | 86.26 | 86.53 | 10.10 | 16.90 | 88.40 | 60.53 |
| TPN (Yang et al., 2020) | CNN | 37.86 | 8.79 | 68.25 | 90.04 | 97.34 | 61.52 |
| TIN (Shao et al., 2020) | CNN | 33.78 | 21.47 | 81.59 | 79.44 | 97.88 | 63.97 |
| TRN (Zhou et al., 2018) | CNN | 38.92 | 26.64 | 91.34 | 93.98 | 93.97 | 71.26 |
| TSM (Lin et al., 2019) | CNN | 70.37 | 54.70 | 78.46 | 70.37 | 96.76 | 76.40 |
| X3D (Feichtenhofer, 2020) | CNN | 92.39 | 37.27 | 65.72 | 49.60 | 97.51 | 77.09 |
| UniFormer V2 (Li et al., 2022a) | TF | 20.05 | 14.81 | 45.21 | 99.21 | 96.89 | 57.55 |
| TimeSformer (Bertasius et al., 2021) | TF | 73.14 | 20.17 | 74.80 | 39.40 | 92.32 | 64.28 |
| VideoSwin (Liu et al., 2022) | TF | 62.29 | 8.01 | 91.82 | 45.83 | **99.29** | 67.27 |
| MViT V2 (Li et al., 2022b) | TF | 76.34 | **98.29** | 47.50 | 96.62 | 97.58 | 79.90 |
| Qwen2.5-VL-7B (Bai et al., 2025) | MLLM | 25.86 | 27.06 | 68.51 | 43.26 | 71.15 | 47.30 |
| GPT-4.1 mini (OpenAI, 2025) | MLLM | 26.07 | 33.78 | 94.07 | 57.19 | 87.64 | 59.62 |
| VidGuard-R1 (CoT) | MLLM | 36.52 | 16.02 | 99.35 | 76.94 | 99.94 | 66.09 |
| VidGuard-R1 (GRPO, GenVideo-pretrained, Zero-shot) | MLLM | 97.24 | 96.59 | 99.88 | 99.93 | 88.14 | 96.37 |
| VidGuard-R1 (GRPO) | MLLM | **97.38** | 94.98 | **99.90** | **99.99** | 95.46 | **97.53** |

## A  ADDITIONAL SETUP

To further guide the model during RL training, we incorporate a length-based reward strategy. We promote informative yet concise reasoning by rewarding outputs that are neither too brief nor excessively long. Specifically, if the model predicts the correct answer and the length of the response falls within the range $[l_{\min}, l_{\max}]$, an additional reward $\omega$ is assigned. Let $l_i$ be the length of the model's response for the $i$-th video. The reward is defined as:

$$
r_i^{total} = \begin{cases} r_i + \omega, & \text{if } o_i \text{ is correct and } l_{\min} \leq l_i \leq l_{\max} \\ r_i, & \text{otherwise} \end{cases}
\tag{4}
$$

where we set $\omega = 0.1$, $l_{\min} = 320$, and $l_{\max} = 512$.

## B  COMPREHENSIVE BENCHMARK EVALUATION

In this section, we provide extended benchmark results for **VidGuard-R1** alongside additional MLLMs. Table 10 presents mean Top-1 accuracy on GenVidBench across multiple video datasets, including CNN and Transformer baselines as well as selected MLLM variants. Table 11 reports comprehensive F1 and recall scores on the GenVideo dataset, including all models provided in the official benchmark alongside our MLLM variants. These extended tables offer a complete comparison of performance across all evaluated models.

## C  ZERO-SHOT GENERALIZATION TO UNSEEN GENERATIVE MODELS

To evaluate the robustness of **VidGuard-R1** beyond the curated training sources, we assess its zero-shot performance on a diverse set of recently released generative video models that were not used during training, including Gen-3 Alpha (run, 2025), Pika (pik, 2025), Pika 2.2 (pik, 2025), Luma Ray2 (lum, 2025), Sora (sor, 2025), Veo2 (veo, 2025), Veo3 (veo, 2025), and Wan 2.1 (wan, 2025). Table 12 summarizes performance across these unseen systems. VidGuard-R1 achieves accuracy above 80% in all cases, reaching up to 96.36%, demonstrating strong generalization to more recent and increasingly realistic generative models.

These analyses demonstrate that **VidGuard-R1** generalizes effectively beyond the curated generative sources and remains robust across a wide range of unseen, high-quality video generation models.

Table 11: Extended GenVideo results with **VidGuard-R1** and additional MLLMs, evaluated by F1 and recall scores

| Model | Detection level | Metric | Sora | Morph Studio | Gen2 | HotShot | Lavie | Show-1 | Moon Valley | Crafter | Model Scope | Wild Scrape | Mean |
|---|---|---|---|---|---|---|---|---|---|---|---|---|---|
| F3Net (Qian et al., 2020) | Image | R | 0.83 | 0.99 | 0.98 | 0.77 | 0.57 | 0.36 | 0.99 | 0.99 | 0.89 | 0.76 | 0.81 |
| | | F1 | 0.50 | 0.94 | 0.96 | 0.81 | 0.69 | 0.49 | 0.93 | 0.96 | 0.88 | 0.82 | 0.80 |
| NPR (Tan et al., 2024) | Image | R | 0.91 | 0.99 | 0.99 | 0.24 | 0.89 | 0.57 | 0.97 | 0.99 | 0.94 | 0.87 | 0.84 |
| | | F1 | 0.27 | 0.84 | 0.91 | 0.30 | 0.86 | 0.59 | 0.81 | 0.91 | 0.81 | 0.81 | 0.71 |
| STIL (Gu et al., 2021) | Video | R | 0.78 | 0.98 | 0.98 | 0.76 | 0.61 | 0.53 | 0.99 | 0.97 | 0.94 | 0.65 | 0.82 |
| | | F1 | 0.38 | 0.90 | 0.94 | 0.78 | 0.72 | 0.62 | 0.90 | 0.94 | 0.88 | 0.72 | 0.78 |
| VideoMAE (Tong et al., 2022) | Video | R | 0.67 | 0.96 | 0.98 | 0.96 | 0.77 | 0.80 | 0.97 | 0.96 | 0.96 | 0.68 | 0.87 |
| | | F1 | 0.62 | 0.95 | 0.98 | **0.96** | 0.86 | 0.87 | 0.96 | 0.97 | **0.96** | 0.79 | 0.89 |
| MINTIME-CLIP (Coccomini et al., 2024) | Video | R | 0.89 | 1.00 | 0.98 | 0.26 | 0.96 | 0.98 | 0.99 | 1.00 | 0.84 | 0.82 | 0.87 |
| | | F1 | 0.49 | 0.93 | 0.96 | 0.37 | 0.94 | 0.92 | 0.92 | 0.96 | 0.84 | 0.85 | 0.82 |
| FTCN-CLIP (Zheng et al., 2021) | Video | R | 0.87 | 1.00 | 0.98 | 0.17 | 0.97 | 0.91 | 1.00 | 1.00 | 0.85 | 0.82 | 0.86 |
| | | F1 | 0.78 | 0.98 | 0.98 | 0.29 | 0.98 | 0.94 | 0.98 | **0.99** | 0.90 | 0.89 | 0.87 |
| TALL (Xu et al., 2023) | Video | R | 0.91 | 0.98 | 0.97 | 0.83 | 0.76 | 0.79 | 0.99 | 0.98 | 0.94 | 0.66 | 0.88 |
| | | F1 | 0.26 | 0.82 | 0.89 | 0.74 | 0.77 | 0.72 | 0.81 | 0.90 | 0.80 | 0.67 | 0.74 |
| CLIP (Radford et al., 2021) | Image | R | 0.94 | 0.99 | 0.91 | 0.77 | 0.88 | 0.86 | 0.99 | 0.99 | 0.84 | 0.84 | 0.90 |
| | | F1 | 0.28 | 0.84 | 0.86 | 0.72 | 0.85 | 0.76 | 0.82 | 0.91 | 0.76 | 0.79 | 0.76 |
| DeMamba-CLIP (Chen et al., 2024a) | Video | R | 0.95 | 1.00 | 0.98 | 0.69 | 0.92 | 0.93 | 1.00 | 1.00 | 0.83 | 0.82 | 0.91 |
| | | F1 | 0.64 | 0.96 | 0.97 | 0.78 | 0.94 | 0.92 | 0.95 | 0.98 | 0.87 | 0.87 | 0.89 |
| XCLIP (Ni et al., 2022) | Video | R | 0.82 | 0.99 | 0.93 | 0.61 | 0.79 | 0.69 | 0.97 | 0.99 | 0.77 | 0.83 | 0.84 |
| | | F1 | 0.31 | 0.88 | 0.90 | 0.65 | 0.82 | 0.70 | 0.86 | 0.93 | 0.75 | 0.82 | 0.76 |
| DeMamba-XCLIP (Chen et al., 2024a) | Video | R | 0.98 | 1.00 | 0.99 | 0.65 | 0.94 | 0.98 | 1.00 | 1.00 | 0.92 | 0.89 | 0.93 |
| | | F1 | 0.64 | 0.96 | 0.97 | 0.75 | **0.95** | 0.95 | 0.95 | 0.97 | 0.92 | **0.91** | 0.90 |
| Qwen2.5-VL-7B (Bai et al., 2025) | MLLM | R | 0.58 | 0.56 | 0.54 | 0.33 | 0.43 | 0.38 | 0.81 | 0.63 | 0.51 | 0.70 | 0.54 |
| | | F1 | 0.74 | 0.72 | 0.70 | 0.49 | 0.60 | 0.55 | 0.90 | 0.77 | 0.68 | 0.82 | 0.70 |
| GPT-4.1 mini (OpenAI, 2025) | MLLM | R | 0.43 | 0.67 | 0.56 | 0.54 | 0.63 | 0.56 | 0.92 | 0.67 | 0.69 | 0.69 | 0.65 |
| | | F1 | 0.60 | 0.80 | 0.72 | 0.70 | 0.77 | 0.72 | 0.96 | 0.80 | 0.82 | 0.82 | 0.72 |
| VidGuard-R1 (CoT) | MLLM | R | 0.92 | 0.89 | 0.91 | 0.90 | 0.98 | 0.79 | 0.99 | 0.85 | 0.89 | 0.87 | 0.90 |
| | | F1 | 0.90 | 0.91 | 0.95 | 0.89 | 0.99 | 0.81 | 0.95 | 0.89 | 0.85 | 0.88 | 0.90 |
| VidGuard-R1 (GRPO, GenVidBench-pretrained, Zero-shot) | MLLM | R | 0.95 | 0.98 | 0.90 | 0.89 | 0.97 | 0.85 | 0.99 | 0.93 | 0.81 | 0.87 | 0.92 |
| | | F1 | 0.93 | 0.93 | 0.96 | 0.91 | 0.99 | 0.82 | 0.95 | 0.89 | 0.85 | 0.88 | 0.91 |
| VidGuard-R1 (GRPO) | MLLM | R | 0.95 | 1.00 | 0.98 | 0.94 | 0.98 | 0.95 | 0.97 | 0.99 | 0.94 | 0.91 | 0.96 |
| | | F1 | **0.97** | **0.99** | **0.99** | 0.91 | **0.99** | 0.89 | **0.99** | **0.99** | 0.95 | 0.90 | **0.96** |

Table 12: Zero-shot detection accuracy on unseen generative models

| Model | Total | Correct | Incorrect | Accuracy (%) |
|---|---|---|---|---|
| Gen-3 Alpha | 56 | 49 | 7 | 87.50 |
| Pika | 110 | 101 | 9 | 91.82 |
| Pika 2.2 | 110 | 106 | 4 | 96.36 |
| Luma Ray2 | 110 | 98 | 12 | 89.09 |
| Sora | 110 | 102 | 8 | 92.73 |
| Veo2 | 52 | 45 | 7 | 86.54 |
| Veo3 | 55 | 45 | 10 | 81.82 |
| Wan2.1 | 55 | 46 | 9 | 83.64 |

## D  PROMPT

Figure 5 shows the base prompt used for the real-vs-fake classification task. Annotators are instructed to assess whether a video is real or AI-generated by analyzing key visual and physical properties.

Figures 6 and 7 provide category-specific rationale collection prompts. In particular, Figure 6 presents the prompt for identifying visual cues of realism in real videos, while Figure 7 focuses on spotting artifacts in AI-generated videos. Both prompts guide annotators to evaluate videos across four diagnostic categories: motion consistency, lighting consistency, texture artifacts, and physics violations.

Figure 8 illustrates the LLM-as-a-judge prompt used to evaluate rationale quality. In this setting, GPT-4.1 mini rates the quality of model-generated explanations on a 1–10 scale, where a score of 10 corresponds to excellent quality and full alignment with the ground truth rationale.

**Prompt for Distinguishing Real from AI-Generated Content**

**SYSTEM:**
A conversation between User and Assistant. The user asks a question, and the Assistant solves it.
The assistant first thinks about the reasoning process in the mind and then provides the user with the
answer. The reasoning process and answer are enclosed within <think> </think> and <answer> </answer>
tags, respectively, i.e., <think> reasoning process here </think><answer> answer here </answer>

**USER:**
<video> Decide whether a video looks a real one or a generated from the AI world model.

Figure 5: Prompt for identifying realism cues in real videos across four categories

**Rationale Collection Prompt for Real Videos**

<video> This is a real-world video. Your task is to provide a detailed guide of which specific parts of the
video should be examined to identify signs of real across four key categories: Motion Consistency, Lighting
Consistency, Texture Artifacts, and Physics Violations. For each category, highlight critical areas or
elements within the video.

Figure 6: Prompt for identifying realism cues in real videos across four categories

**Rationale Collection Prompt for AI-Generated Videos**

<video> This video has been generated by an AI model. Your task is to provide a detailed guide on which
parts of the video identify signs of generation across four key categories: Motion Consistency, Lighting
Consistency, Texture Artifacts, and Physics Violations. For each category, highlight critical areas or
elements within the video.

Figure 7: Prompt for identifying artifacts in AI-generated videos across four categories

# E    CASE STUDIES ON EXPLANATIONS

## E.1    GENVIDBENCH

Figures 9–12 present inference examples for videos synthesized by four distinct AI models included in the GenVidBench testing dataset: MuseV (Xia et al., 2024), SVD (Blattmann et al., 2023), CogVideo (Hong et al., 2022), and Mora (Yuan et al., 2024).

## E.2    GENVIDEO

Figures 13–22 show inference examples for videos generated by ten different AI models included in the GenVideo testing dataset: Sora (sor, 2025), Morph Studio (mor, 2025), Gen2 (Esser et al., 2023), HotShot (hot, 2025), Lavie (Wang et al., 2025), Show-1 (Zhang et al., 2024a), Moonvalley (moo, 2025), Crafter (Chen et al., 2023), ModelScope (Wang et al., 2023a), and DreamVideo (Wei et al., 2024).

**LLM-as-a-Judge Prompt for Rationale Quality Evaluation in Real vs. Generated Video Classification**

**SYSTEM:**
You are an expert judge evaluating the **explanation quality** of a vision-language model (VLM) that decides whether a video is real or AI-generated. The model outputs a binary decision (**real** or **fake**) along with a **rationale** explaining the basis of its decision.
The rationale should focus on four visual diagnostic categories:
Your evaluation should consider the following five criteria:
**Accuracy**: Does the rationale identify the key generation artifacts or natural signals relevant to the decision?
**Alignment with Ground Truth**: Does the rationale emphasize the same visual evidence?
**Specificity**: Is the rationale grounded in the visual content (i.e., free from hallucinations or vague generalities)?
**Conciseness**: Is the rationale clearly and efficiently worded without unnecessary length or redundancy?
**Consistency with Model's Answer**: Is the rationale logically coherent with the model's predicted label (real/fake)? For example, if the model said "fake," does the rationale convincingly support this?
Please provide your evaluation in the following format:
Start with "ANALYSIS:" and write a concise paragraph analyzing the rationale with respect to the five criteria.
Then, on the next line, write "SCORE:" followed by a single integer from 1 to 10 (10 = excellent quality and full alignment).

**USER:**
Let's evaluate the model's rationale!
**Ground truth answer (real/fake)**:
{ground truth answer}
**Ground truth rationale**:
{ground truth rationale}
**Model answer (real/fake)**:
{model answer}
**Model rationale**:
{model rationale}

Figure 8: LLM-as-a-judge prompt for rationale quality evaluation

**Question:** Does this video look real or generated by an AI world model?     Real or **Generated**

```
<think>Let me think about this. When evaluating whether this video is real or AI-generated, I'll consider the four factors
mentioned: Motion Consistency, Lighting Consistency, Texture Artifacts, and Physics Violations.
1. **Motion Consistency**: In the video, there are no moving objects that would allow me to assess motion consistency. The
scene appears static, with no visible movement of ships, boats, or any other elements. This lack of motion makes it
difficult to determine if the video is real or AI-generated based on motion consistency alone.
2. **Lighting Consistency**: The lighting in the video appears to be consistent with what one would expect in a real-world
scenario. The shadows and highlights on the buildings and water reflect natural lighting conditions. However, without more
context or a comparison to other videos, it's hard to definitively say if the lighting is perfectly consistent with real-
world conditions.
3. **Texture Artifacts**: Upon closer inspection, the textures of the buildings, water, and other elements appear somewhat
uniform and lack the fine detail and variation one would expect in a real-world photograph. This could be a result of the AI
generation process, where textures might not be perfectly realistic.
4. **Physics Violations**: There are no obvious physics violations in this scene. The positioning of the buildings, water,
and boats seems plausible and follows the expected physics of a harbor or cityscape. However, the lack of motion and the
uniformity of textures could still be indicative of AI generation.
Considering these points, the video does not exhibit clear motion, and the textures and lighting are somewhat uniform, which
could suggest AI generation. However, the absence of motion and the lack of fine detail make it challenging to definitively
conclude.
In conclusion, while the video does not exhibit clear motion or physics violations, the uniformity of textures and the lack
of fine detail suggest that it could be AI-generated. However, more context or a comparison to other videos would be needed
for a more definitive assessment.</think>
<answer> Generated </answer>
```

Figure 9: VidGuard-R1 (GRPO) inference on a MuseV-generated video

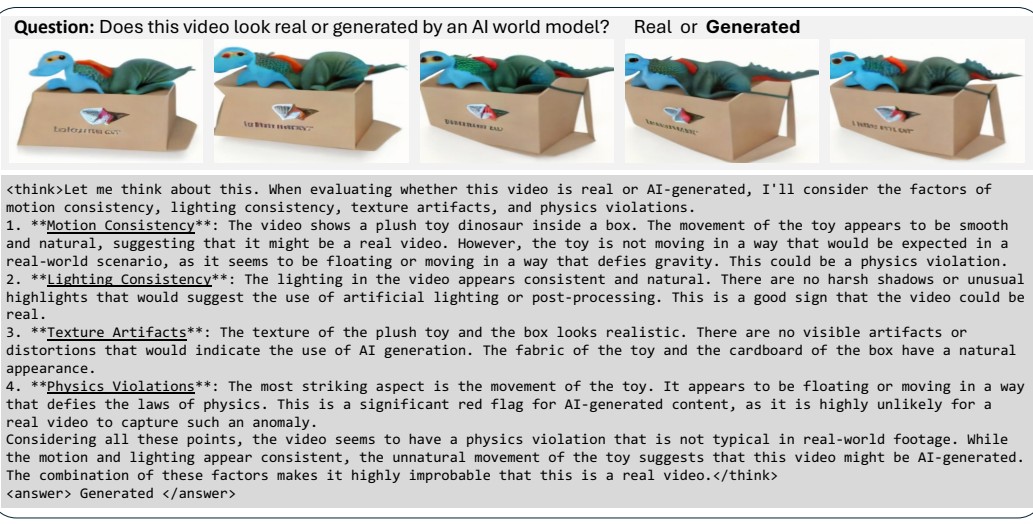

**Question:** Does this video look real or generated by an AI world model?    Real  or  **Generated**

<think>Let me think about this video carefully. The scene appears to be a chaotic event, possibly a protest or a police intervention, with people in motion. As I analyze the video, I'll consider the factors mentioned: Motion Consistency, Lighting Consistency, Texture Artifacts, and Physics Violations.
1. **Motion Consistency**: The individuals in the video seem to be moving in a way that is somewhat realistic. However, the speed and direction of the movement appear slightly inconsistent at times. This could be due to the AI's attempt to simulate human movement, but the slight discrepancies might indicate a lack of perfect synchronization.
2. **Lighting Consistency**: The lighting in the video appears to be consistent with natural daylight. There are no sudden changes in brightness or shadows that would suggest artificial lighting. The shadows and highlights seem to follow the expected patterns for an outdoor scene.
3. **Texture Artifacts**: Upon closer inspection, there are some subtle artifacts in the textures of the clothing and the environment. The edges of objects and people seem a bit jagged, which is a common issue with AI-generated images. This could be due to the AI's attempt to render textures in a realistic manner, but the artifacts are noticeable.
4. **Physics Violations**: There are no obvious physics violations in the video. The people and objects are moving in a way that is plausible for a real-world scenario. The interactions between the individuals and the environment also seem natural. Considering these observations, the video does exhibit some inconsistencies that are typical of AI-generated content. The motion and texture artifacts are subtle but present, and the overall scene is consistent with what one might expect from a real-world event. However, the slight discrepancies in motion consistency and the presence of texture artifacts suggest that this video is likely generated by an AI model rather than a real recording.
In conclusion, while the video appears to be a realistic depiction of a chaotic event, the subtle inconsistencies in motion and texture artifacts point towards it being an AI-generated video.</think>
<answer> Generated </answer>

Figure 10: VidGuard-R1 (GRPO) inference on a SVD-generated video

**Question:** Does this video look real or generated by an AI world model?    Real  or  **Generated**

<think>Let me think about this. When evaluating whether this video is real or AI-generated, I'll consider the factors of motion consistency, lighting consistency, texture artifacts, and physics violations.
1. **Motion Consistency**: The video shows a plush toy dinosaur inside a box. The movement of the toy appears to be smooth and natural, suggesting that it might be a real video. However, the toy is not moving in a way that would be expected in a real-world scenario, as it seems to be floating or moving in a way that defies gravity. This could be a physics violation.
2. **Lighting Consistency**: The lighting in the video appears consistent and natural. There are no harsh shadows or unusual highlights that would suggest the use of artificial lighting or post-processing. This is a good sign that the video could be real.
3. **Texture Artifacts**: The texture of the plush toy and the box looks realistic. There are no visible artifacts or distortions that would indicate the use of AI generation. The fabric of the toy and the cardboard of the box have a natural appearance.
4. **Physics Violations**: The most striking aspect is the movement of the toy. It appears to be floating or moving in a way that defies the laws of physics. This is a significant red flag for AI-generated content, as it is highly unlikely for a real video to capture such an anomaly.
Considering all these points, the video seems to have a physics violation that is not typical in real-world footage. While the motion and lighting appear consistent, the unnatural movement of the toy suggests that this video might be AI-generated. The combination of these factors makes it highly improbable that this is a real video.</think>
<answer> Generated </answer>

Figure 11: VidGuard-R1 (GRPO) inference on a CogVideo-generated video

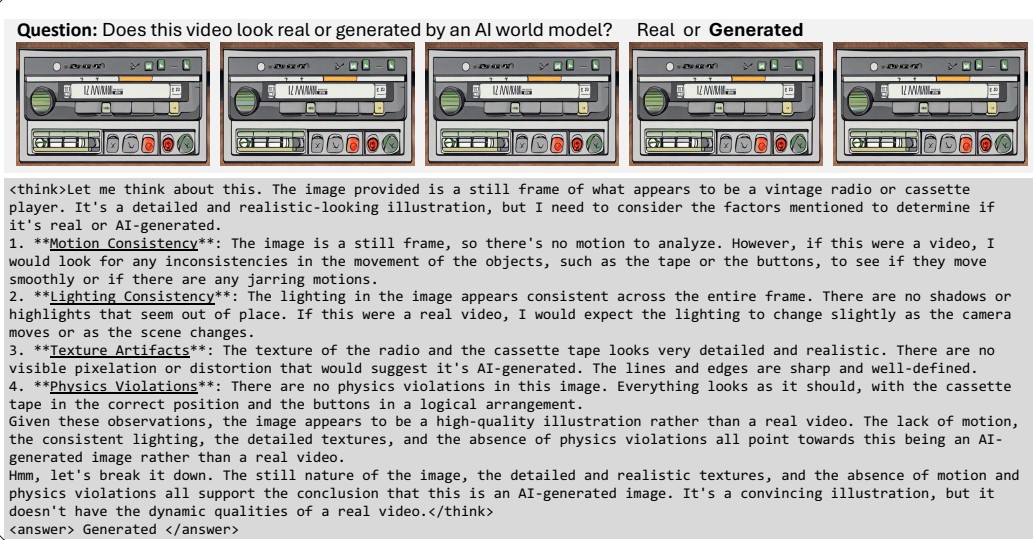

Figure 12: VidGuard-R1 (GRPO) inference on a Mora-generated video

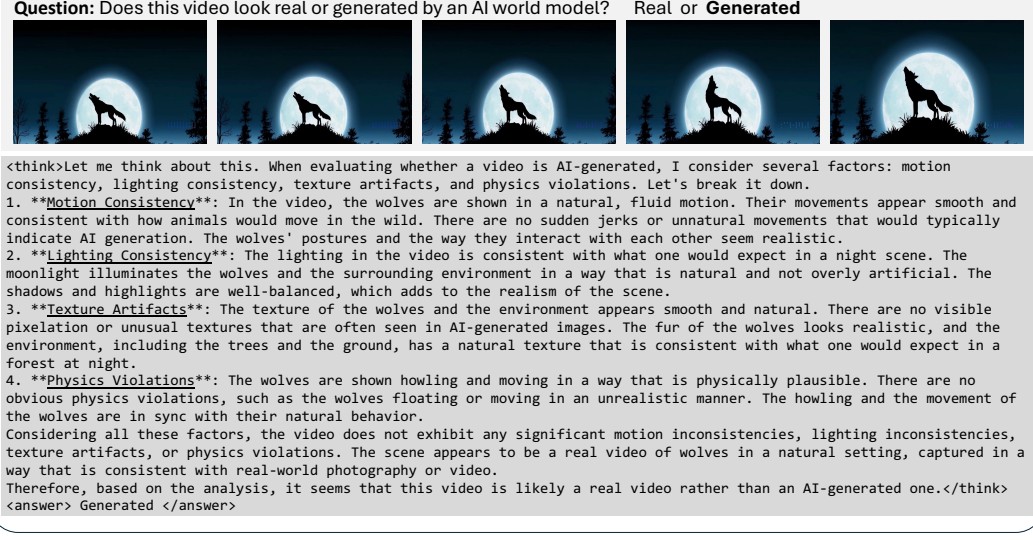

Figure 13: VidGuard-R1 (GRPO) inference on a Sora-generated video

**Question:** Does this video look real or generated by an AI world model?     Real  or  **Generated**

<think> Let me think about this. When evaluating whether this video is real or AI-generated, I'll consider the four factors you mentioned: Motion Consistency, Lighting Consistency, Texture Artifacts, and Physics Violations.
1. **Motion Consistency**: In the video, people are walking, and their movements appear somewhat natural. However, the speed and fluidity of their movements might not be entirely consistent with real-world human motion. AI-generated videos often have a slightly artificial feel to the movement, which can be subtle but noticeable.
2. **Lighting Consistency**: The lighting in the video seems to be consistent with what you'd expect in a real-world setting. The shadows and highlights on the people and buildings appear natural, suggesting that the lighting was carefully simulated to match real-world conditions.
3. **Texture Artifacts**: Upon closer inspection, the textures of the people's clothing and the environment (like the cobblestones and building facades) appear somewhat uniform and lack the fine detail and variation you'd typically see in a real photograph or video. This could be a sign that the textures are generated by an AI model.
4. **Physics Violations**: There are no obvious physics violations in this video. The people are walking in a way that is consistent with how they would move in a real-world scenario. The shadows and reflections also align with what you'd expect in a real environment.
Considering these points, the video seems to have some artificial qualities, particularly in the texture and motion consistency. While it's not definitively clear if this is an AI-generated video, the subtle inconsistencies in texture and motion suggest that it might be. However, without more advanced analysis tools, it's challenging to be certain. The video could still be a high-quality real-world recording that has been edited or enhanced in some way.</think>
<answer> Generated </answer>

Figure 14: VidGuard-R1 (GRPO) inference on a Morph Studio-generated video

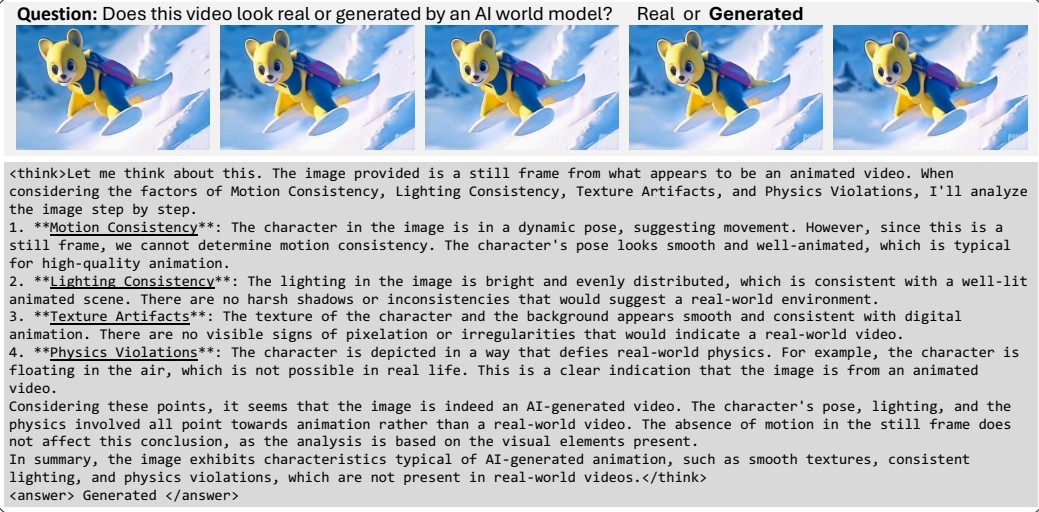

**Question:** Does this video look real or generated by an AI world model?     Real  or  **Generated**

<think>Let me think about this. The image provided is a still frame from what appears to be an animated video. When considering the factors of Motion Consistency, Lighting Consistency, Texture Artifacts, and Physics Violations, I'll analyze the image step by step.
1. **Motion Consistency**: The character in the image is in a dynamic pose, suggesting movement. However, since this is a still frame, we cannot determine motion consistency. The character's pose looks smooth and well-animated, which is typical for high-quality animation.
2. **Lighting Consistency**: The lighting in the image is bright and evenly distributed, which is consistent with a well-lit animated scene. There are no harsh shadows or inconsistencies that would suggest a real-world environment.
3. **Texture Artifacts**: The texture of the character and the background appears smooth and consistent with digital animation. There are no visible signs of pixelation or irregularities that would indicate a real-world video.
4. **Physics Violations**: The character is depicted in a way that defies real-world physics. For example, the character is floating in the air, which is not possible in real life. This is a clear indication that the image is from an animated video.
Considering these points, it seems that the image is indeed an AI-generated video. The character's pose, lighting, and the physics involved all point towards animation rather than a real-world video. The absence of motion in the still frame does not affect this conclusion, as the analysis is based on the visual elements present.
In summary, the image exhibits characteristics typical of AI-generated animation, such as smooth textures, consistent lighting, and physics violations, which are not present in real-world videos.</think>
<answer> Generated </answer>

Figure 15: VidGuard-R1 (GRPO) inference on a Gen2-generated video

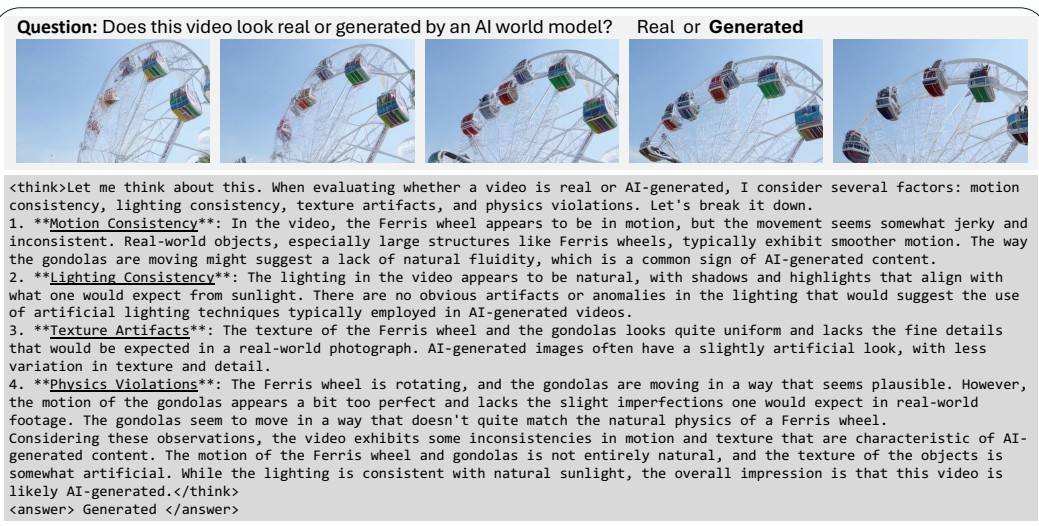

Figure 16: VidGuard-R1 (GRPO) inference on a HotShot-generated video

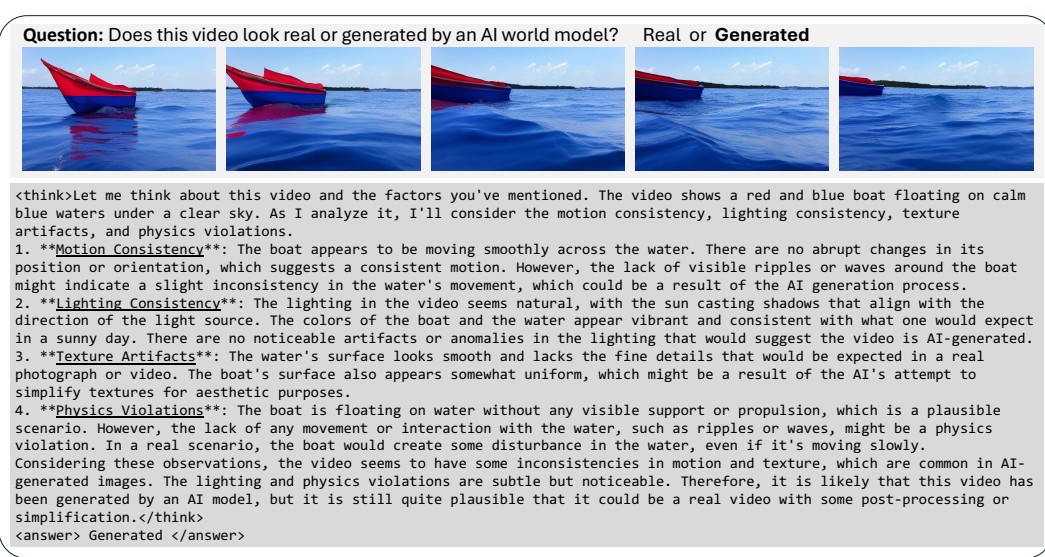

Figure 17: VidGuard-R1 (GRPO) inference on a Lavie-generated video

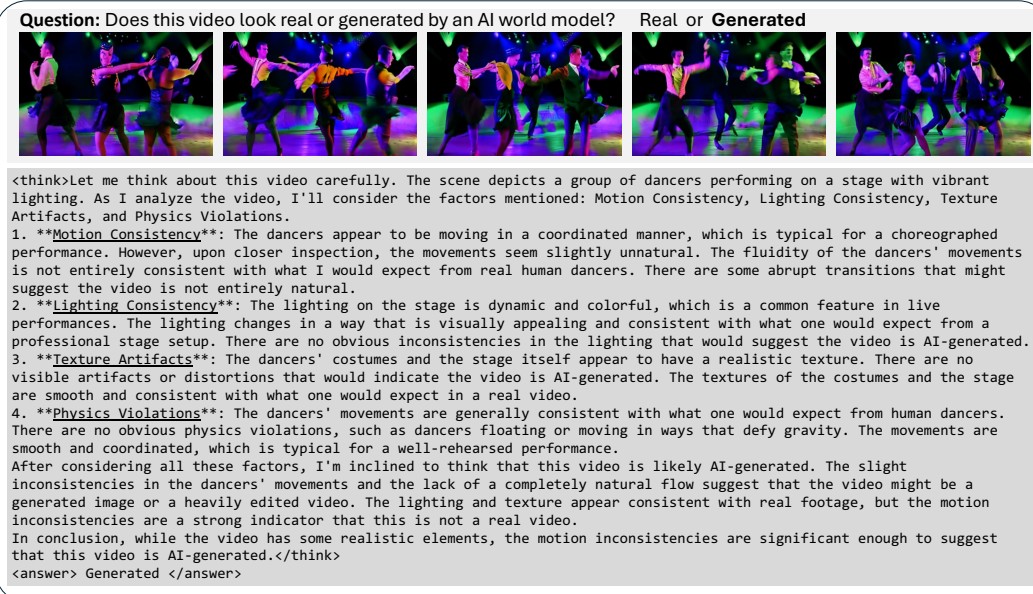

Figure 18: VidGuard-R1 (GRPO) inference on a Show-1-generated video

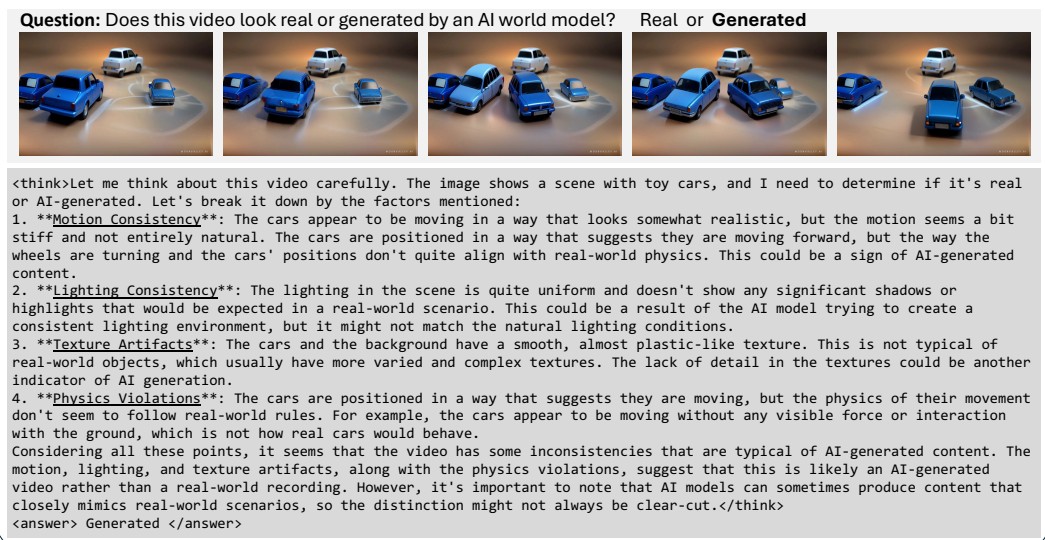

Figure 19: VidGuard-R1 (GRPO) inference on a Moonvalley-generated video

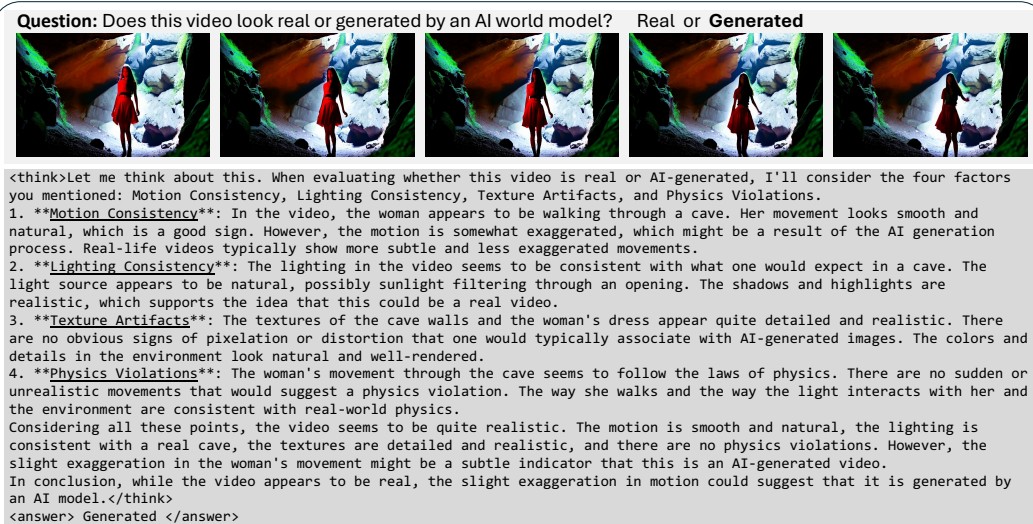

Figure 20: VidGuard-R1 (GRPO) inference on a Crafter-generated video

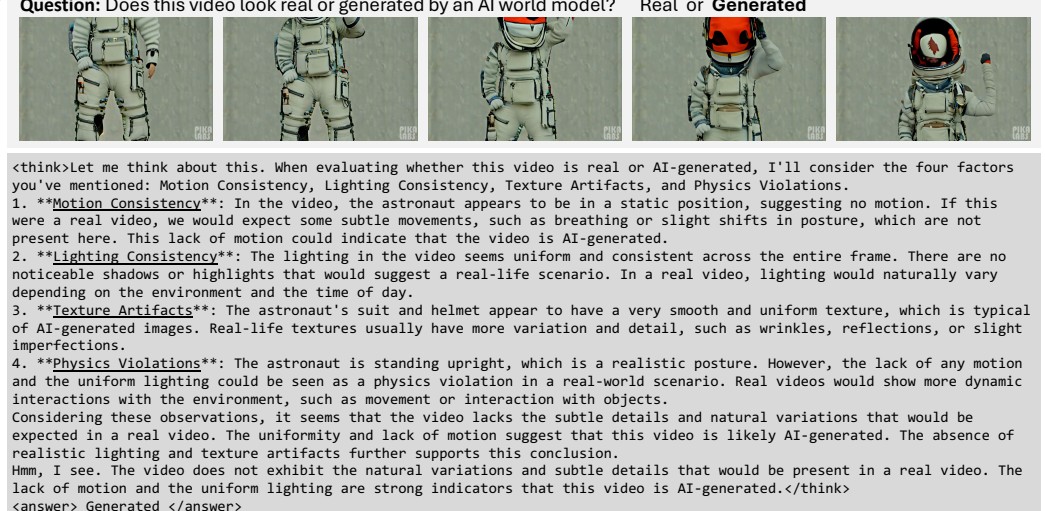

Figure 21: VidGuard-R1 (GRPO) inference on a ModelScope-generated video

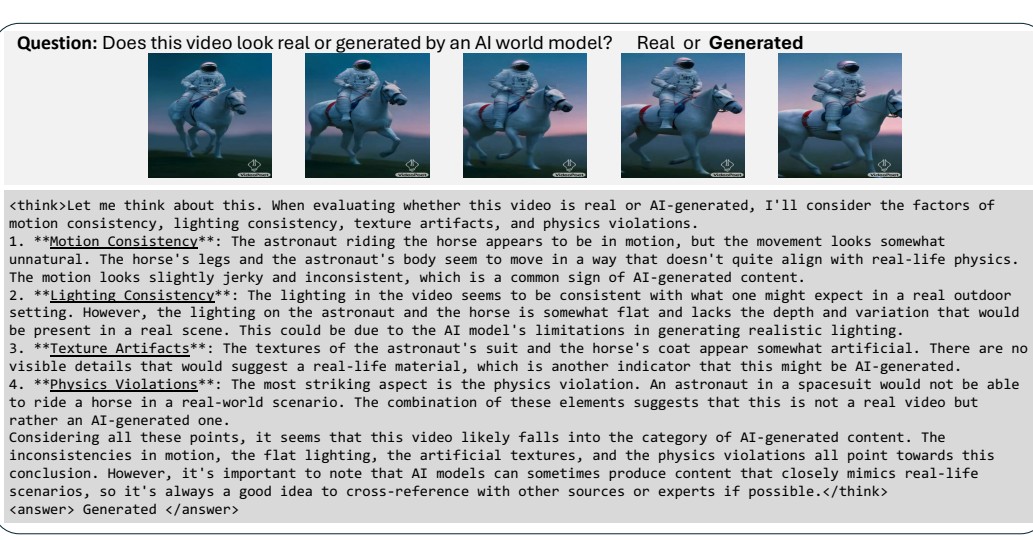

Figure 22: VidGuard-R1 (GRPO) inference on a DreamVideo-generated video

