# OpenReview forum: "VidGuard-R1: AI-Generated Video Detection and Explanation via Reasoning MLLMs and RL"
_ICLR.cc/2026/Conference — ICLR 2026 Poster_

### Official Review · Reviewer_3kcH · 2025-10-20

**Soundness:** 3
**Presentation:** 3
**Contribution:** 2
**Rating:** 4
**Confidence:** 4

**Summary:**

The paper introduces VidGuard-R1, a multimodal large language model (MLLM)-based system designed for detecting AI-generated videos and providing interpretable explanations for its decisions. The model utilizes reinforcement learning (RL) and group relative policy optimization (GRPO) to fine-tune a Qwen-VL model for better accuracy and reasoning capability. VidGuard-R1 targets temporal artifacts and generation complexity to improve detection performance. It achieves high accuracy on several benchmarks and presents detailed case studies showing its ability to generate interpretable rationales behind predictions.

**Strengths:**

1. Dataset Scale: The paper introduces a dataset of 140k real and AI-generated videos designed to challenge current detection methods.
2. Introduce Explanation in AIGV Detection: the ability to explain why a video is classified as real or fake.
3. Advanced Fine-Tuning Techniques (GRPO and RL): The use of GRPO (Group Relative Policy Optimization) and reinforcement learning for fine-tuning is a notable strength. The GRPO-TA and GRPO-Q variants further enhance performance by targeting temporal artifacts and generation complexity, offering additional levels of refinement that set VidGuard-R1 apart from previous methods.
4. Paper Clarity: The paper presents a clear and structured methodology for VidGuard-R1, and the results are presented clearly in tables and figures that allow for easy comparison of VidGuard-R1’s performance against other methods.

**Weaknesses:**

1. Dataset Generalization: While the paper introduces a well-curated dataset, its generalizability could be questioned, as it only includes videos generated by HunyuanVideo and CogVideoX. The paper does not explore the performance of models when trained on the proposed dataset and evaluated on others.
2. Model Generalization: the lack of cross-dataset validation raises concerns about the model’s ability to generalize to unseen data or models outside of these two specific generative systems. The paper does not explore the performance of VidGuard-R1 when trained on one dataset and evaluated on others, or trained on some AIGV model generations and tested on other unseen AIGV model generations.
3. Ground-Truth Labeling Reliability: The method of conditioning CoT rationales on ground-truth labels from a more powerful model may compromise the genuineness of the model's discrimination ability. Since the annotations are directly influenced by the 72B model’s reasoning, the 7B model might struggle to perform true reasoning on its own, leading to potential overfitting to the provided cues rather than developing independent judgment.
4. Innovation in the framework: Existing works such as Q-insight and VQ-insight have already employed similar reinforcement learning-based approaches GRPO for optimizing Qwen2.5-VL 7B models and improving their reasoning abilities. The main difference in this paper seems to be the use of a new reward method or a new dataset rather than introducing a fundamentally new approach or technique.
5. Limited Explanation Comparison and Model Comparison: The explanation comparison fails to include a more powerful model like Qwen2.5-VL 72B, GPT5, Grok, etc., which is crucial for a more comprehensive evaluation of the explanation quality.

**Questions:**

1. Have you tested the model on unseen AI-generated video models? How does the performance of VidGuard-R1 compare when it is trained on AIGV model-generated videos from one source and tested on videos generated from other unseen AI models?
2. Could you provide more details on how VidGuard-R1 performs when trained on your proposed dataset but evaluated on other datasets or with video content generated by different AI models?
3. Given that the CoT rationales are conditioned on ground-truth labels from a more powerful model (Qwen2.5-VL 72B), how do you ensure that VidGuard-R1 can develop independent reasoning without overfitting to the potentially biased or predefined cues provided by the larger model?
4. Given that similar approaches, such as Q-insight and VQ-insight, have used GRPO for optimizing multi-modal models, can you clarify the novel contributions of your work, especially in relation to GRPO? How does the reward method or the new dataset significantly advance the existing literature?
5. Could you provide a more comprehensive comparison of VidGuard-R1’s explanation quality against state-of-the-art models (such as GPT-5 or Qwen2.5-VL 72B) to give a clearer understanding of how VidGuard-R1’s interpretability holds up against larger, more capable models?

---

> ### Author Response · Authors · 2025-11-20
>
> We thank the reviewer for the constructive feedback and for recognizing the strengths of our dataset scale, explanation capability, GRPO-based fine-tuning, and overall clarity. Below, we address each concern in detail.
>
> ---
>
> ### *W1: Dataset Generalization*
>
> We include dataset-mixing experiments to show cross-dataset complementarity:
>
> #### **Results**
> | Test Dataset | Training Source               | Accuracy (%) |
> |--------------|-------------------------------|--------------|
> | Ours         | VidGuard-R1 (Ours only)       | 81.65        |
> | Ours         | VidGuard-R1 (Ours + GenVideo) | 82.97        |
> | GenVideo     | VidGuard-R1 (GenVideo only)   | 97.53        |
> | GenVideo     | VidGuard-R1 (Ours + GenVideo) | 97.98        |
>
> Even though the models used to generate videos differ between datasets, **mixing them still improves accuracy**.
> Moreover, as shown in **W2**, a model trained on our dataset using HunyuanVideo and CogVideoX also performs strongly on videos generated by many *unseen* AIGV models.
>
> We appreciate the reviewer’s observation regarding dataset limitations. We acknowledge that our current dataset uses videos generated by only **HunyuanVideo** and **CogVideoX**. However, the **purpose** of proposing this dataset is to highlight a critical issue in existing benchmarks:
>
>
> ### **Existing benchmarks contain major shortcuts**
>
> For example:
>
> #### **GenVidBench**
> | Video Source         | Type | Task | Resolution | FPS |
> |------------------|------|------|------------|-----|
> | Vript             | Real | –    | 1280×720   | 30  |
> | HD-VG-130M     | Real | –    | 1280×720   | 30  |
> | **—**            | **—** | **—** | **—**      | **—** |
> | Pika             | Fake | T2V  | 1088×560   | 24  |
> | VideoCrafter2      | Fake | T2V  | 512×320    | 10  |
>
> #### **GenVideo**
> | Video Source  | Type | Task | Time  | Resolution | FPS | Length |
> |---------------|------|------|-------|------------|-----|--------|
> | Kinetics-400  | Real | –    | 17.05 | 224–340    | –   | 5–10s  |
> | Youku-mPLUG   | Real | –    | 23.07 | –          | –   | 10–120s|
> | ZeroScope     | Fake | T2V  | 23.07 | 1024×576   | 8   | 3s     |
> | SVD           | Fake | I2V  | 23.12 | 1024×576   | 8   | 4s     |
>
> As shown, **real vs. fake videos differ drastically in resolution, FPS, bitrate, and data sources**, creating strong shortcuts that inflate performance.
>
>
> ### **Our dataset eliminates shortcuts**
> Our dataset enforces:
> - identical **bitrate**, **frame rate**, and **resolution**
> - **fake videos using image/caption extracted from their corresponding real videos**
>   → ensuring similar distribution
>
> Thus, our dataset **complements** (rather than replaces) GenVidBench/GenVideo by providing a more realistic evaluation environment.
>
> We acknowledge the limited model coverage. At the time of dataset construction, **HunyuanVideo** and **CogVideoX** were the primary open-source models that supported **joint text–image–conditioned** video generation; other diffusion models offered only text- or image-only conditioning.
>
> ---
> ### *W2: Model Generalization* & *Q1: Unseen AI-generated video models?* &  *Q2: Details on how VidGuard-R1 performs when trained on your proposed dataset but evaluated on other datasets or models?*
>
>
> To evaluate robustness against unseen synthesis methods, we trained VidGuard-R1 on our proposed dataset and conducted **zero-shot testing** on videos generated by completely **unseen** AIGV models:
>
> #### **Zero-shot Evaluation on Unseen Generators**
> | Model        | Total | Correct | Accuracy (%) |
> |--------------|-------|---------|--------------|
> | Gen-3 Alpha  | 56    | 49      | 87.50        |
> | HunyuanVideo | 110   | 101     | 91.82        |
> | Pika         | 110   | 101     | 91.82        |
> | Pika 2.2     | 110   | 106     | **96.36**    |
> | Luma Ray2    | 110   | 98      | 89.09        |
> | Sora         | 110   | 102     | 92.73        |
> | Veo2         | 52    | 45      | 86.54        |
> | Veo3         | 55    | 45      | 81.82        |
> | Wan2.1       | 55    | 46      | 83.64        |
>
> VidGuard-R1 consistently achieves **80–96% accuracy**, frequently surpassing **90%**, showing that the model captures **generator-agnostic cues** rather than memorizing artifacts.
>
> VidGuard-R1 also shows strong zero-shot generalization on:
> - **GenVidBench** (Table 2)
> - **GenVideo** (Table 3)
>
> To further strengthen the analysis, we added **cross-domain OOD evaluations** using mismatched training/testing splits.
>
>
> ### **GenVideo OOD Setup**
> - Real: *Kinetics-400, Youku-mPLUG → MSR-VTT*
> - Synthetic: *10 training generators → 10 unseen testing generators*
>
> ### **GenVidBench OOD Setup**
> - Real: *Vript → HD-VG-130M*
> - Synthetic:
>   *Training:* Pika, VC2, ModelScope, T2V-Zero
>   *Testing:* MuseV, SVD, Mora, CogVideo
>
> ### **Result**
> Across both benchmarks, **VidGuard-R1 achieves >96% OOD accuracy**, demonstrating robustness against:
> - cross-domain shifts
> - cross-generator shifts
> - unseen model families and styles

---

> ### Author Response · Authors · 2025-11-20
>
> ### *W3 & Q3: Does distilling CoT from a stronger model harm independent reasoning? Overfitting to teacher cues?*
> Using CoT generated by a stronger model to bootstrap SFT before RL is a common and well-established practice, and has been shown effective in many prior works. Typically, a more capable model or an agent system first produces a set of CoT rationales, which are then validated, filtered, or refined via rejection sampling before being used to conduct SFT on the student model, enabling long-form CoT capability.
>
> For example, in **Thyme** [1], Qwen-72B is used to generate code-based tool-calling CoT for **Qwen2.5-VL 7B**, followed by SFT and RL. The resulting student model achieves generalization and SOTA performance, with highly convincing reasoning traces. Similarly, **R1-OneVision** [2] and **Open Vision Reasoner** [3] both use **DeepSeek-R1 (671B)** to generate CoT for Qwen2.5-VL 7B during SFT, then apply RL on math and visual reasoning tasks. These works consistently demonstrate that distilling CoT from stronger models yields substantial gains, achieving much better reasoning quality than training from scratch. In **ReasonGen-R1** [4], GPT-4o–constructed CoT is used to train autoregressive image-generation reasoning, and ablations confirm that this rational learning is indispensable.
>
> We would like to emphasize that the SFT stage is **not** responsible for teaching the final reasoning logic; rather, it serves to **expand the exploration space** before RL and improve RL efficiency. Learning from a stronger model provides a shortcut for acquiring a reasonable reasoning format and cognitive behavior. Therefore, we argue that the assertion that *“a 7B model might struggle to perform true reasoning on its own”* is not accurate. True reasoning ability is acquired during the **RL stage**, through policy-gradient optimization not during SFT.  The responsibility of SFT is simply to provide a broad, high-quality exploration prior, and a stronger teacher greatly accelerates this process.
>
> Large models such as Qwen-72B primarily provide **richer and more diverse CoT samples**. Similar reasoning traces can also be generated by prompting the student model itself to “think step by step,” followed by rejection sampling, as in **Seed 1.5-VL** [5]. We respectfully ask the reviewer to clarify why distillation would inherently lead to “overfitting to biased or predefined cues.” When the training schedule is moderate, distillation does not cause collapse; for instance, **DeepSeek-R1-Distill-Qwen-7B** [6] is trained purely with SFT on teacher CoT and still demonstrates strong independent reasoning ability.
>
> Our results also indicate that VidGuard-R1 does **not** overfit. We perform only **one epoch** of CoT SFT, yet reinforcement learning yields substantial gains thereafter:
>
> * **GenVidBench:** 66.09 → **97.53**
> * **HunyuanVideo:** 63.19 → **86.17**
>
> Since RL uses **no teacher model**, these improvements demonstrate that the student model maintains and further develops its own reasoning capability. If distillation had harmed independence, RL diversity and rewards would collapse, which we do not observe.
>
> Finally, both the **Detection Accuracy** (W5 & Q5) and **Human Ranking** results support that our gains reflect **true improvements**, not shortcut exploitation. Frontier models such as GPT-4o and Qwen2.5-VL-72B remain below 60% in zero-shot detection, whereas VidGuard-R1 surpasses **80% accuracy** and receives the **most human-aligned rationale rankings**. This confirms that label-guided SFT combined with RL-based reasoning offers meaningful, non-overfit benefits.
>
>
> ### **References**
> [1] Zhang, Yifan et al. “Thyme: Think Beyond Images.” ArXiv abs/2508.11630 (2025).
> [2] Yang, Yi et al. “R1-Onevision: Advancing Generalized Multimodal Reasoning through Cross-Modal Formalization.” ArXiv abs/2503.10615 (2025).
> [3] Wei, Yana et al. “Open Vision Reasoner.” ArXiv abs/2507.05255 (2025).
> [4] Zhang, Yu et al. “ReasonGen-R1.” ArXiv abs/2505.24875 (2025).
> [5] Guo, Dong et al. “Seed1.5-VL Technical Report.” ArXiv abs/2505.07062 (2025).
> [6] DeepSeek AI. “DeepSeek-R1-Distill-Qwen-7B.” HuggingFace (2025).

---

> ### Author Response · Authors · 2025-11-20
>
> ### *W4 & Q4: Novelty beyond Q-Insight / VQ-Insight? What is fundamentally new in our GRPO framework?*
> Our contributions are novel in **application**, **technical design**, and **benchmark construction**, and differ substantively from Q-Insight and VQ-Insight.
>
> #### **1. A new application paradigm: reasoning-based video authenticity detection**
> We present the first system enabling an MLLM to:
> - detect whether a video is AI-generated, **and**
> - explain *why* it is fake via CoT.
>
> This supports **human-in-the-loop safety**, essential for mitigating AI-generated misinformation.
> Prior GRPO works do *not* address this domain, nor provide explainable video authenticity reasoning.
>
> #### **2. First to activate MLLM reasoning for video authenticity using SFT + RL**
> To our knowledge, ours is the **first** framework applying:
> - MLLM + CoT-SFT cold-start
> - domain-aware GRPO
> - reasoning-before-classification paradigm
>
> for video authenticity detection.
>
> #### **3. Technical innovations unique to this application**
> These components do not appear in Q-Insight or VQ-Insight:
>
> ##### **(a) A rigorously controlled dataset with no shortcuts**
> We equalize:
> - bitrate
> - FPS
> - resolution
> - content semantics (fake videos use captions extracted from the real videos)
>
> This forces models to rely on **reasoning**, not dataset artifacts.
>
> ##### **(b) GRPO-Q: Diffusion-step–aware quality rewards**
> Diffusion steps encode how far synthetic videos deviate from real distribution, providing a meaningful prior for RL.
>
> ##### **(c) GRPO-TA: Motion-consistency–aware temporal rewards**
> We explicitly penalize the model from relying solely on **spatial artifacts**, enforcing temporal reasoning—critical and unique to video.
>
> These designs address challenges that Q-Insight/VQ-Insight do not face.
>
> #### **4. Significant performance and interpretability gains**
> The combination of dataset design + reasoning-based RL yields:
> - substantial improvements in fake-video detection
> - strong zero-shot generalization
> - human-aligned, content-grounded rationales
>
> We will expand related-work comparison accordingly in the revised version.

---

> ### Author Response · Authors · 2025-11-21
>
> ### *W5: Limited Explanation Comparison and Model Comparison: The explanation comparison fails to include a more powerful model like Qwen2.5-VL 72B, GPT5, Grok, etc., which is crucial for a more comprehensive evaluation of the explanation quality.*
> ### *&*
> ### *Q5: Could you provide a more comprehensive comparison of VidGuard-R1’s explanation quality against state-of-the-art models (such as GPT-5 or Qwen2.5-VL-72B) to better understand how VidGuard-R1’s interpretability compares to larger, more capable models?*
>
> We sincerely thank the reviewers for their insightful suggestions. In response, we have added two human evaluations to more thoroughly assess the quality of VidGuard-R1’s explanations and compare them against state-of-the-art models.
>
> We conducted a human evaluation measuring the quality of VidGuard-R1’s explanations:
>
> - **Annotators:** 5 evaluators
> - **Samples:** 20 correctly identified fake videos
> - **Tasks:**
>   1. **Yes/No:** Is the rationale consistent with the prediction?
>   2. **Quality score (0–5):** Does it accurately reflect the video content?
>
> ### **Results**
> - **Alignment:** 89% agreement
> - **Average score:** **3.9 / 5**
> - **Low-score cases:** 2–3 videos where the model referenced subtle artifacts that were not obvious to humans
>
> These results show that **VidGuard-R1 generally produces coherent and human-aligned rationales**, with occasional reliance on very subtle cues that humans may miss.
>
> ---
>
> To compare rationale quality against **GPT-4o** and **Qwen2.5-VL-72B**, we further conducted a **blind human ranking study**:
>
> - **Participants:** 5
> - **Assigned videos:** Each participant viewed **10 real** + **10 fake** videos
> - **Procedure:**
>   For each video, participants were given three explanations (from VidGuard-R1, GPT-4o, Qwen2.5-VL-72B) **without knowing the source model**, and asked to rank them **1 (best), 2, 3 (worst)** based on explanation quality and accuracy.
>
> ### **Human Ranking Results**
>
> | Model                | #Rank-1 | #Rank-2 | #Rank-3 | Avg. Rank |
> |----------------------|---------|---------|---------|-----------|
> | **VidGuard-R1 (GRPO)** | **49**     | 35      | 16      | **1.67**      |
> | GPT-4o               | 30      | 23      | 38      | 2.08      |
> | Qwen2.5-VL-72B       | 21      | 42      | 46      | 2.22      |
>
> VidGuard-R1 clearly produces explanations that **align better with human perception and reasoning** compared to much larger frontier MLLMs.
>
> ---
>
> We additionally extended Table 1 in the main paper by evaluating **stronger MLLMs** (GPT-4o, Qwen2.5-VL-72B) on our proposed dataset in **AI-generated video detection**.
>
> ### **Detection Accuracy on Our Dataset**
>
> | Method / Model            | Type | CogVideoX | HunyuanVideo |
> |---------------------------|-------|-----------|--------------|
> | Qwen2.5-VL-7B             | MLLM  | 50.95     | 52.83        |
> | Qwen2.5-VL-72B            | MLLM  | 54.17     | 55.82        |
> | GPT-4.1-mini              | MLLM  | 54.95     | 55.31        |
> | GPT-4o                    | MLLM  | 56.81     | 57.42        |
> | **VidGuard-R1 (GRPO)**    | MLLM  | **81.30** | **81.90**    |
>
> These results demonstrate that **VidGuard-R1 trained with SFT + RL on top of Qwen2.5-VL-7B is significantly more effective** for AI-generated video forensics. Existing frontier MLLMs, when used in a zero-shot manner, **struggle with this task**, underscoring the practical value and contribution of our method.

---

> > ### Author Response · Authors · 2025-11-27
> >
> > We appreciate your thorough and helpful critique. We believe the clarifications and new experimental results we've included successfully address all the issues you identified and provide a better understanding of our work. Should any aspect still be unclear, we are ready to offer more details during the discussion. We are grateful for your valuable input and hope our responses merit a re-evaluation of the current score.

---

> > > ### Comment · Reviewer_3kcH · 2025-11-28
> > >
> > > I appreciate the author's responses, which have addressed all my concerns. I will raise my score, and I hope that this work will contribute to the advancement of this field.

---

> ### Author Response · Authors · 2025-11-28
>
> Thank you very much for your thoughtful review and feedback. We are glad that our responses addressed your concerns, and we sincerely appreciate your decision to raise the score. Should you have any further suggestions or comments during the discussion period, we would be happy to take them into consideration.

---

### Official Review · Reviewer_1fnv · 2025-10-29

**Soundness:** 3
**Presentation:** 4
**Contribution:** 3
**Rating:** 6
**Confidence:** 3

**Summary:**

This paper presents VidGuard-R1, a multi-modal large language model (MLLM)–based system for AI-generated video detection and explanation. The work addresses growing societal risks from realistic video generation by developing a reasoning-capable authenticity detector that produces both accurate judgments and interpretable explanations. The model fine-tunes Qwen-VL using a two-stage framework: supervised chain-of-thought (CoT) initialization followed by reinforcement learning with Group Relative Policy Optimization (GRPO) and two specialized reward models—one emphasizing temporal artifacts (GRPO-TA) and another focusing on generation complexity (GRPO-Q). The authors construct a large dataset of real and synthetic videos generated by recent diffusion-based models and demonstrate that VidGuard-R1 achieves strong zero-shot and fine-tuned performance on several benchmarks, accompanied by qualitative examples illustrating its reasoning process.

**Strengths:**

- The paper tackles an important societal problem by adapting reasoning MLLMs for deepfake and synthetic video detection.
- The combination of supervised CoT initialization and RL-based GRPO fine-tuning is conceptually sound and clearly explained, with reward models thoughtfully designed to encourage temporal and quality-aware reasoning.
- The curated dataset is large, diverse, and standardized to minimize shortcut cues such as duration or resolution differences, making it well suited for robust learning. The dataset can be of good value to the community as well.
- Experimental results show consistent and often state-of-the-art accuracy across multiple benchmarks, including GenVideo and GenVidBench, with competitive zero-shot generalization.
- The inclusion of reasoning traces and qualitative explanations enhances transparency and potential user trust, distinguishing this work from prior black-box detectors.

- The paper is well written and easy to follow, with clear explanations of both the methodology and its motivation. Figures and examples  effectively convey the reasoning process and highlight the interpretability of the model’s predictions.

**Weaknesses:**

- Although interpretability is a stated goal, the paper does not evaluate the quality of explanations beyond qualitative examples. It remains unclear whether the rationales are causally consistent with correct predictions or merely plausible-sounding. Established metrics (e.g., LLM-as-judge, human evaluation, or coherence scores) should be adopted to substantiate this claim.

- Several key ideas have been explored in prior work on image and video authenticity detection (e.g., SafeWatch, DeMamba-XCLIP, or recent surveys on AI-generated media detection). The paper could more clearly articulate what is fundamentally new in its contribution beyond integrating existing elements.

- The related work section omits relevant prior literature on video safety and explainable detection, such as SafeWatch (ICLR 2025) and other text-to-video safety models, which share similar objectives and techniques. A more explicit comparison—methodologically and empirically—would strengthen positioning.

- The training dataset, while large, is largely synthesized from one or two generative sources (e.g., HunyuanVideo-I2V, CogVideoX). This raises concerns about generalization to unseen generation models. Evaluating across broader generative distributions or conducting cross-model tests would better support claims of robustness.

**Questions:**

- The statement that “directly assigning a reward of 1 to real videos and 0 to fake ones presents challenges” is not justified theoretically or empirically; a short ablation or analysis on this design choice would clarify its necessity.

- Will the dataset and code be publicly released for community use?

---

> ### Author Response · Authors · 2025-11-20
>
> We thank the reviewer for the constructive and detailed feedback, as well as for recognizing the strengths of our methodology, dataset construction, and presentation quality. We address each concern point-by-point below.
>
> ---
>
> ### *W1: Although interpretability is a stated goal, the paper does not evaluate the quality of explanations beyond qualitative examples. It remains unclear whether the rationales are causally consistent with correct predictions or merely plausible-sounding. Established metrics (e.g., LLM-as-judge, human evaluation, or coherence scores) should be adopted to substantiate this claim.*
>
> We agree that evaluating the quality of VidGuard-R1’s Chain-of-Thought (CoT) rationales is essential for substantiating interpretability claims. To complement the qualitative examples in the original submission, we conducted a **human evaluation study** of VidGuard-R1 (GRPO-Q):
>
> **Participants**
> - 5 annotators
>
> **Procedure**
> We randomly sampled 20 **correctly classified fake videos**. For each video, annotators:
> 1. Assessed whether the CoT rationale was aligned with the model’s “fake” decision (**Yes/No**)
> 2. Rated the rationale quality on a **0–5 scale**, evaluating clarity, grounding, and justification strength
>
> **Results**
> - **Yes/No alignment:** 89% average agreement (1 example unanimously judged insufficient)
> - **Quality ratings:** Average **3.9/5**, indicating strong correspondence between rationales and video content
> - **Low-scoring outliers:**
>   - 2–3 rationales received ≤2
>   - These typically relied on subtle visual cues not easily perceptible to human annotators
>   - Importantly, final predictions were correct, but the rationales referenced non-human-visible evidence
>
> **Interpretation**
> VidGuard-R1 generally produces rationales consistent with human expectations, while occasionally relying on imperceptible cues.
> We emphasize that we **do not** claim CoT explanations are always correct. Even with ~90% alignment, hallucinations remain possible. Therefore, CoT explanations should be presented to users with proper disclaimers about the inherent uncertainties of AI-generated reasoning.
>
> However, the practical value lies in enabling **human-in-the-loop verification**. By exposing interpretable cues, users can more confidently assess authenticity and mitigate risks associated with misleading AI-generated content.
>
> As a future direction, we plan to perform **human-in-the-loop experiments** to verify whether adding interpretability improves human accuracy and speed in AI-generated video detection.
>
> ---
>
> ### *W2 & W3: Prior work such as SafeWatch, DeMamba-XCLIP… what is fundamentally new? Related work omissions.*
>
> We appreciate the suggestion to better contrast our work with prior literature, including SafeWatch, DeMamba-XCLIP, and surveys on AI-generated video detection.
>
> It is important to highlight that **VidGuard-R1 is the first system to introduce multimodal LLMs into AI-generated video detection**, and the first to use **SFT + RL (GRPO)** to enable the model to *generate end-to-end reasoning* explaining why a video is real or fake.
>
> Key differentiators include:
>
> - **First SFT+RL pipeline for reasoning-based AI video authenticity system with multimodal LLM**
>   VidGuard-R1 uses SFT+RL to enable multimodal LLMs' reasoning ability to produce *explicit* rationales for authenticity, rather than relying on concept filters or policy compliance.
>
> - **Novel RL algorithms (GRPO-Q and GRPO-TA)**
>   These introduce quality-aware and temporal-consistency-aware rewards, demonstrating measurable performance gains.
>
> - **A new, shortcut-free benchmark**
>   Unlike existing benchmarks (e.g., GenVideo, GenVidBench) that exhibit strong shortcuts in resolution, FPS, bitrate, or data source imbalance, our dataset enforces strict equivalence in:
>   - bitrate
>   - resolution
>   - frame rate
>   - content distribution (fake videos generated from real-video captions)
>
> Compared to SafeWatch and similar works:
> - SafeWatch focuses on **policy compliance** and **unsafe content detection** (e.g., erotic, violent, censored concepts)
> - Our task focuses on **authenticity** and **fake feature detection**
> - The motivations, supervision signals, and evaluation setups are fundamentally different
>
> We will revise the related work section to highlight both methodological and conceptual differences, as well as empirical improvements driven by reasoning-centric training.

---

> ### Author Response · Authors · 2025-11-20
>
> ### *W4: Dataset uses only HunyuanVideo and CogVideoX — concerns about generalization*
>
> We appreciate the reviewer’s observation. While our dataset is currently generated from two models, the purpose of our benchmark is to eliminate shortcuts widely present in existing datasets.
>
> #### **Existing benchmarks contain large-scale shortcuts**
> Real/fake videos from GenVidBench and GenVideo show strong mismatches in: resolution, FPS, bitrate, and underlying content sources
>
> These shortcuts substantially inflate detector performance.
>
> #### **Our dataset eliminates shortcuts by design**
> We enforce:
> - identical bitrate, frame rate, and resolution
> - fake videos generated from captions extracted from the corresponding real videos
>
> This produces a far more realistic and fair detection setting.
>
> #### **Generalization experiments**
>
> We include dataset-mixing experiments showing complementary performance and cross-domain benefits:
>
> | Test Set   | Training Source               | Acc (%) |
> |------------|-------------------------------|---------|
> | Ours       | Ours only                     | 81.65   |
> | Ours       | Ours + GenVideo               | 82.97   |
> | GenVideo   | GenVideo only                 | 97.53   |
> | GenVideo   | Ours + GenVideo               | 97.98   |
>
> Mixing data improves accuracy on all datasets and demonstrates a cross-model improvement.
>
> #### **Zero-shot generalization to unseen models**
> We also tested on several *recent generative models* unseen during training:
>
> | Model        | Total | Correct | Acc (%) |
> |--------------|-------|---------|---------|
> | Gen-3 Alpha  | 56    | 49      | 87.50   |
> | HunyuanVideo | 110   | 101     | 91.82   |
> | Pika         | 110   | 101     | 91.82   |
> | Pika 2.2     | 110   | 106     | 96.36   |
> | Luma Ray2    | 110   | 98      | 89.09   |
> | Sora         | 110   | 102     | 92.73   |
> | Veo2         | 52    | 45      | 86.54   |
> | Veo3         | 55    | 45      | 81.82   |
> | Wan 2.1      | 55    | 46      | 83.64   |
>
> VidGuard-R1 consistently maintains **>80%** accuracy, with a best of **96.36%**, showing strong robustness to new, more advanced generative models.
>
> VidGuard-R1 already shows strong zero-shot generalization on:
> - **GenVidBench** (Table 2)
> - **GenVideo** (Table 3)
>
> These benchmarks perform **cross-domain OOD evaluations** with mismatched training/testing splits.
>
> ##### **GenVideo OOD setup**
> - Real: *Kinetics-400, Youku-mPLUG → MSR-VTT*
> - Synthetic: 10 training generators → 10 unseen testing generators
>
> ##### **GenVidBench OOD setup**
> - Real: *Vript → HD-VG-130M*
> - Synthetic: *Pika, VC2, ModelScope, T2V-Zero → MuseV, SVD, Mora, CogVideo*
>
> #### **Cross-domain OOD evaluation**
> Across both GenVideo and GenVidBench OOD splits, VidGuard-R1 achieves **>96%** accuracy, demonstrating robust resistance to:
> - domain shifts
> - generator shifts
> - unseen generative distributions
>
> These results show that our reasoning-enhanced training significantly improves *true generalization*, not exploitation of shortcuts.

---

> ### Author Response · Authors · 2025-11-21
>
> ### *Q1: Assign reward 1 to real and 0 to fake?*
>
> This design applies **only** to GRPO-Q and GRPO-TA. These two variants are auxiliary tasks intended to *support* the standard GRPO objective by providing additional supervision that encourages the model to generate correct CoT explanations for real vs. fake classification.
>
> For the **standard GRPO setting**, directly using a **binary reward** is indeed the most appropriate approach. This follows the classic formulation of **Reinforcement Learning with Verifiable Rewards (RLVR)**, where policy gradients reinforce the sampling probability of CoT traces and answers associated with reward = 1. No special reward design is needed in this case.
>
> The reason we introduced non-binary rewards is:
>
> - **GRPO-Q:**
>   Fake videos generated at different diffusion steps have varying levels of realism. A binary reward would collapse these distinctions, so different scores are necessary to reflect the gradation of quality.
>
> - **GRPO-TA:**
>   Temporally perturbed videos lie *between* real and AI-generated videos. Assigning them the same 0/1 reward as the standard GRPO setting would produce inconsistent learning signals and degrade temporal-reasoning behavior.
>
> For a more detailed discussion of the motivations behind GRPO-Q and GRPO-TA, please refer to our responses to Reviewer **AX7C Q2 and Q3**.
>
> ---
>
>
> ### *Q2: publicly released*
>
> We commit to fully open-sourcing all training and evaluation code, the dataset, annotation scripts, and our trained models to support community development. We look forward to broader collaboration and discussion, and hope this work can contribute meaningfully to the safe development of AI.

---

> > ### Author Response · Authors · 2025-11-27
> >
> > Thank you for your positive and thoughtful comments. We appreciate your careful reading of our work, and we hope that our clarifications and additional experiments further strengthen the points you raised and provide a more complete picture of our contributions. If any aspects would benefit from additional explanation, we would be happy to elaborate during the discussion period. We sincerely value your constructive feedback and hope that our responses support a favorable reassessment where appropriate.

---

### Official Review · Reviewer_AX7C · 2025-10-29

**Soundness:** 3
**Presentation:** 3
**Contribution:** 3
**Rating:** 4
**Confidence:** 4

**Summary:**

The authors propose a detector finetuning MLLM using GRPO producing 2 reward models, one based on temporal artifacts and the other focused on image generation quality. They also produced an additional dataset with standardised formatting of the videos so as to prevent models from taking shortcuts to easily identify real and fake videos. The authors also produce CoT data that supports the production of text that can be used as a possible explanation.

**Strengths:**

S1: The way the finetuning procedure is performed is non-trivial and proposes interesting ideas. The methodology is simple but effective.

S2: The way the dataset is constructed, and the special attention in avoiding models taking shortcuts is appreciated.

S3: The availability of rich CoT data could be really impactful and useful for many applications, but ...

**Weaknesses:**

W1: The way the CoT data is generated is likely to produce hallucinations. The models may easily base their explanations based on prior knowledge about video artifacts, and the model could be correct at random.

W2: Claims about interpretability should be made cautiously: the generated text is simply a sequence of tokens, not really an explanation of the model's decision.

**Questions:**

Q1: Could the authors discuss or introduce quality checks to ensure the CoT data is high quality?

Q2: The idea of using the number of diffusion steps to control the quality is interesting, have you considered also manipulating other hyperparameters (e.g. changing the number of frames?)

Q3: Is there a specific rationale for the hcoice of those 2 specific temporal artifacts? Indeed they're quite different from naturally occurring ones. Is there a chance that applying more realistic ones could result in even better performance?

In relation to W1,W2,Q1: It would be interesting if the authors could show 1) human studies to assess the quality of the explanations, 2) produce a study on how many times the explanations are in disagreement with humans and whether the disagreement is caused by an hallucination or by the fact humans leverage different cues. These would also guarantee the produced CoT data is high quality and can be more broadly useful for the community.

I am very favourable to increasing my score if these concerns are addressed, as I think it will greatly increase the impact of the author's work.

---

> ### Author Response · Authors · 2025-11-20
>
> We sincerely thank the reviewer for the positive and constructive feedback on our methodology and dataset design. Below, we provide detailed clarifications and additional analyses addressing each point raised.
>
> ---
>
> ### *W1: The way the CoT data is generated is likely to produce hallucinations… & Q1: Could the authors discuss or introduce quality checks?*
>
> We appreciate the reviewer’s concern regarding potential hallucinations in CoT outputs and agree that interpretability should be treated carefully. To strengthen our claims, we performed a **human evaluation study** focused on assessing the alignment and quality of VidGuard-R1’s rationales.
>
> **Study Setup**
>
> - **Annotators:** 5 evaluators
> - **Samples:** 20 randomly selected, correctly identified *fake* videos
> - **Tasks:**
>   1. Judge whether the rationale is consistent with the predicted label (Yes/No)
>   2. Provide a 0–5 quality score after reviewing the corresponding video
>
> **Results**
>
> - **Rationale–decision alignment:** 89% average agreement
> - **Rationale quality:** Average score of **3.9/5**
> - **Outliers:**
>   - 2–3 rationales received ≤2
>   - These cases typically referenced subtle texture cues not easily perceivable to humans
>   - Importantly, the **predictions were correct**, but the explanations relied on imperceptible artifacts
>
> These findings indicate that VidGuard-R1 generally produces **coherent and contextually grounded CoT rationales**, while also revealing limitations when relying on subtle cues. This human study will be included in the revised version to transparently present both strengths and limitations of CoT-based interpretability.
>
> ---
>
> ### *W2: Claims about interpretability should be made cautiously: the generated text is simply a sequence of tokens, not really an explanation of the model's decision.*
>
> Please refer to our response to **W1 & Q1** for CoT-quality evaluation. Here, we provide additional clarification.
>
> We appreciate the reviewer’s reminder that interpretability claims must be cautious, as generative models indeed may hallucinate, and CoT explanations should not be naively interpreted as definitive evidence. Beyond improving detection accuracy via reasoning-based training, our interpretability contribution lies in proposing a **new interaction paradigm** for AI-based video authenticity assessment—one in which the system provides **reasoning-based explanations** to enhance user understanding and trust.
>
> We emphasize:
>
> - We **do not** claim the generated CoT is always correct.
> - Even though our user study shows ~90% alignment, hallucinations still exist.
> - Therefore, the CoT should be presented with the proper disclaimer that it is an AI-generated interpretation.
>
> However, the value lies in enabling **human-in-the-loop verification**: when users are presented with interpretable cues, they can more reliably judge whether a video is AI-generated. We plan to perform further experiments evaluating whether interpretability helps humans judge AI-generated videos **more accurately and more quickly**, further demonstrating the real-world utility of this contribution.
>
> ---
>
> ### *Q2: Using diffusion steps to control quality is interesting—what about other hyperparameters?*
>
> Thank you for the insightful question. Diffusion models generate samples by solving a reverse-time SDE from noise toward the true data distribution. More diffusion steps provide a trajectory that has a higher likelihood of reaching regions close to the data manifold—an approximation aligned with our motivation to simulate different authenticity levels.
>
> In contrast, modifying factors such as **number of frames**, **resolution**, or **FPS** reflect *video metadata* rather than intrinsic realism. Low-resolution or low-FPS videos can still be fully authentic and natural, which conflicts with detecting **less realistic** AI-generated videos rather than **low-quality** videos. Moreover, these changes also affect the **number of video tokens** used as input for MLLM models, potentially introducing confounding factors. For this reason, we chose the **number of diffusion steps** for our problem.

---

> ### Author Response · Authors · 2025-11-21
>
> ### *Q3: Rationale behind choosing the 2 specific temporal artifacts? Would more realistic ones help?*
>
> Although the two temporal artifacts we use—**reverse** and **repeat**—are not common in real videos, they serve a distinct purpose as a **temporal-reasoning regularizer**, not as attempts to mimic real-world distortions.
>
> Key rationale:
>
> - These artifacts keep **all spatial content untouched**, preventing the model from relying solely on appearance-based shortcuts.
> - They force the model to reason about **motion continuity** and **causal temporal order**—aspects detectors frequently ignore because spatial cues are often easier to exploit.
> - Without such supervision, the model may overfit to spatial artifacts and fail when temporal reasoning is required.
>
> We also chose reverse/repeat because they are lightweight, clean, and easy to apply. More realistic temporal corruptions could be explored in future work, but these simple artifacts already provide strong, stable supervisory signals and yield consistent performance improvements.

---

> > ### Author Response · Authors · 2025-11-27
> >
> > We greatly appreciate your thoughtful and constructive review. We trust that the clarifications and additional experiments we have included adequately resolve the concerns you highlighted and provide a clearer picture of our contributions. Should any aspects remain unclear or merit further elaboration, we would be happy to address them during the discussion phase. Your insightful feedback is sincerely valued, and we hope our responses encourage a reconsideration of the current score where appropriate.

---

### Official Review · Reviewer_acmo · 2025-11-01

**Soundness:** 3
**Presentation:** 2
**Contribution:** 2
**Rating:** 4
**Confidence:** 4

**Summary:**

This paper introduces VIDGuard-R1, a video authenticity detector leveraging multi-modal large language models (MLLMs) and reinforcement learning with group relative policy optimization (GRPO). The approach involves fine-tuning Qwen-VL via supervised and reinforcement learning with two reward models: one designed to incentivize temporal artifact detection and another to guide the model towards assessing generation quality using diffusion step progression. The authors also curate a challenging dataset of 140k real and AI-generated videos. Extensive experiments show VIDGuard-R1 outperforms prior methods and baselines on multiple benchmarks and provides interpretable, step-by-step rationales for its decisions.

**Strengths:**

1. The paper advances the state-of-the-art by developing the first MLLM-based, RL-enhanced video authenticity detector that not only improves accuracy but also generates detailed, interpretable reasoning, as supported by the architecture overview in Figure 1 and illustrated in Figures 3 and 4 showing stepwise explanations.

2.  Introduction of specialized reward models one that encourages careful temporal reasoning (GRPO-TA) and another that uses diffusion step quality signals (GRPO-Q) is a thoughtful step toward aligning model incentives with the challenges of real/fake video discrimination. The mathematical formulations are clearly presented, especially the explicit conditions and reward assignment logic.

3. The authors curate a large, carefully controlled real/fake video dataset (140k paired samples) where low-level cues and trivial metadata are equalized, thus demonstrating scientific rigor in experimental setup. The commitment to removing “shortcut” biases strengthens the validity of their results.

**Weaknesses:**

1. The article proposes a method for identifying fake videos using an enhanced reasoning model and incorporates additional video-related designs in the reward scheme. Although it cannot be ignored the author's contribution of applying enhanced reasoning training to a specific field, the overall approach still follows common training techniques, limiting its overall innovativeness.

2. While the curated dataset is a step forward, its scope is limited by only including videos generated by HunyuanVideo and CogVideoX. As acknowledged in Section 5.1, this restricts the external validity and robustness of the detector. The lack of inclusion of generative outputs from other recent or diverse models (such as those evaluated in the GenVideo or GenVidBench test sets) in the training phase could lead to overfitting to specific generation artifacts. The importance of this limitation is amplified by the single-source pairing and should be further addressed experimentally.

3. The use of reward thresholds and hyperparameters, for example $\alpha_1$, $\alpha_2$, and $\mu$ in Equation (GRPO-TA, Section 3.3.2) or step mapping function $g(\cdot,\cdot)$ in GRPO-Q (Section 3.3.3), is not sufficiently justified theoretically nor by ablation for practical robustness. While Table 5 presents some grid search results for $\alpha_1$ and $\alpha_2$, the rationale for selecting these ranges and assessing sensitivity is shallow—potentially undermining reproducibility if the optimal region is narrow.

**Questions:**

1. How is the detection capability for ood videos?

---

> ### Author Response · Authors · 2025-11-20
>
> We sincerely thank the reviewer for the thoughtful and detailed assessment. We appreciate the recognition of our reward design, dataset construction, and interpretable reasoning pipeline. Below, we address each concern raised by the reviewer:
>
> ---
>
> ### _W1. The article proposes a method for identifying fake videos using an enhanced reasoning model and incorporates additional video-related designs in the reward scheme. Although it cannot be ignored the author's contribution of applying enhanced reasoning training to a specific field, the overall approach still follows common training techniques, limiting its overall innovativeness._
>
> We introduce a method for detecting fake videos using an enhanced reasoning model, supported by video-specific reward designs. While we acknowledge that the overall training pipeline shares similarities with existing reasoning-enhanced LLM approaches, we would like to clarify the **actual contributions** and **novelty** of our work in this specific domain.
>
> Our contributions are as follows:
>
> 1. **A new application paradigm for video authenticity detection.**
>    We are the first to introduce a *reasoning model that explains* why a video is real or fake. This significantly improves user trust and transparency, telling users *why* a video is AI-generated is crucial for enabling **human-in-the-loop oversight**, allowing humans to verify, interpret, and act upon the model’s predictions. Such human-in-the-loop feedback helps mitigate misinformation, supports responsible AI use, and reduces real-world risks associated with AI-generated content. We would like to emphasize that this direction is both meaningful and important.
>
> 2. **Reasoning-then-judgment improves detection accuracy.**
>    Our experiments demonstrate that asking the model to reason first and then make a prediction significantly improves detection performance. Although the concept of eliciting better reasoning was not invented in this work, **we are the first to validate its effectiveness in video authenticity detection**.
>
> 3. **Technical contributions beyond application.**
>    - We propose a *more challenging* dataset.
>    - We introduce **GRPO-TA** and **GRPO-Q**, both of which demonstrably boost RL effectiveness for video detection tasks.
>
> Therefore, we believe our work provides **meaningful contributions** and **domain-specific innovativeness** to both the community and industry.

---

> ### Author Response · Authors · 2025-11-20
>
> ### _W2. While the curated dataset is a step forward, its scope is limited by only including videos generated by HunyuanVideo and CogVideoX. As acknowledged in Section 5.1, this restricts the external validity and robustness of the detector. The lack of inclusion of generative outputs from other recent or diverse models (such as those evaluated in the GenVideo or GenVidBench test sets) in the training phase could lead to overfitting to specific generation artifacts. The importance of this limitation is amplified by the single-source pairing and should be further addressed experimentally._
>
>
>
> We appreciate the reviewer’s observation regarding the dataset limitations. We fully acknowledge that our current dataset uses videos generated by only **HunyuanVideo** and **CogVideoX**. However, the **purpose** of proposing this dataset is to highlight an important limitation in existing benchmarks:
>
> ### Existing benchmarks contain major shortcuts
> For example:
>
> #### **GenVidBench**
> | Video Source                 | Year    | Pair  | Type  | Task | Resolution | FPS | Numbers |
> |------------------------------|---------|-------|--------|------|------------|-----|---------|
> | Vript                        | 2024.04 | –     | Real   | –    | 1280×720   | 30  | 20131   |
> | HD-VG-130M                   | 2023.05 | Pair2 | Real   | –    | 1280×720   | 30  | 13800   |
> | **—**                        | **—**   | **—** | **—**  | **—** | **—**      | **—** | **—** |
> | Pika                         | 2022.05 | Pair1 | Fake   | T2V  | 1088×560   | 24  | 13500   |
> | VideoCrafter2                | 2024.01 | Pair1 | Fake   | T2V  | 512×320    | 10  | 13500   |
>
>
> #### **GenVideo**
> | Video Source       | Type  | Task | Time   | Resolution | FPS | Length | Numbers |
> |--------------------|--------|------|--------|------------|-----|---------|---------------|
> | Kinetics-400       | Real   | –    | 17.05 | 224–340    | –   | 5–10s   | 260,232       |
> | Youku-mPLUG        | Real   | –    | 23.07 | –          | –   | 10–120s | 953,279       |
> | **—**                        | **—**   | **—** | **—**  | **—** | **—**      | **—** | **—** |
> | ZeroScope          | Fake   | T2V  | 23.07 | 1024×576   | 8   | 3s      | 133,169       |
> | SVD                | Fake   | I2V  | 23.12 | 1024×576   | 8   | 4s      | 149,026       |
>
> As shown, **real vs. fake videos in these datasets differ significantly in resolution, FPS, bitrate, and content sources**, which introduces strong shortcuts.
>
> ### Our dataset eliminates shortcuts
> Our dataset strictly enforces:
> - identical **bitrate**
> - identical **frame rate**
> - identical **resolution**
> - fake videos are generated using *captions extracted from the corresponding real videos*
>   → ensuring the prompts match the real content distribution
>
> Thus, our dataset is not meant to replace GenVidBench/GenVideo, but to **complement them** by providing a more realistic evaluation environment.
>
> We acknowledge the limited number of generative models included. At the time of dataset construction, **HunyuanVideo** and **CogVideoX** were the main open-source models supporting large-scale **text–image joint conditioning**, whereas other diffusion models supported only text- or image-only inputs, so we focused on them.
>
>
> ---
>
> ### Addressing *amplified by the single-source pairing*
>
> We include dataset-mixing experiments to show cross-dataset complementarity:
>
> #### **Results**
> | Test Dataset | Training Source              | Accuracy (%) |
> |--------------|------------------------------|---------------|
> | Ours         | VidGuard-R1 (Ours only)      | 81.65         |
> | Ours         | VidGuard-R1 (Ours + GenVideo)| 82.97         |
> | GenVideo     | VidGuard-R1 (GenVideo only)  | 97.53         |
> | GenVideo     | VidGuard-R1 (Ours + GenVideo)| 97.98         |
>
> Even though the models used to generate videos differ between datasets, mixing them **still improves accuracy**, showing the model is *not* overfitting.
>
> ---
>
> ### Zero-shot evaluation on unseen generative models
>
> We further performed zero-shot testing on several recently released models **not used in training**:
>
> | Model        | Total | Correct | Incorrect | Accuracy (%) |
> |--------------|-------|---------|-----------|---------------|
> | Gen-3 Alpha  | 56    | 49      | 7         | 87.50         |
> | HunyuanVideo | 110   | 101     | 9         | 91.82         |
> | Pika         | 110   | 101     | 9         | 91.82         |
> | Pika 2.2     | 110   | 106     | 4         | 96.36         |
> | Luma Ray2    | 110   | 98      | 12        | 89.09         |
> | Sora         | 110   | 102     | 8         | 92.73         |
> | Veo2         | 52    | 45      | 7         | 86.54         |
> | Veo3         | 55    | 45      | 10        | 81.82         |
> | Wan2.1       | 55    | 46      | 9         | 83.64         |
>
> VidGuard-R1 consistently maintains **>80%** accuracy with a peak of **96.36%**, demonstrating strong generalization to **newer, more realistic** generative models.

---

> ### Author Response · Authors · 2025-11-20
>
> ### _W3. The use of reward thresholds and hyperparameters, and in Equation (GRPO-TA, Section 3.3.2) or step mapping function in GRPO-Q (Section 3.3.3), is not sufficiently justified theoretically nor by ablation for practical robustness. While Table 5 presents some grid search results for and, the rationale for selecting these ranges and assessing sensitivity is shallow—potentially undermining reproducibility if the optimal region is narrow._
>
> We appreciate the reviewer’s thoughtful feedback regarding the use of reward thresholds and hyperparameters. We would like to clarify that the original submission includes ablation studies for
>   - α₁, α₂ (GRPO-TA)  (Table 5)
>   - diffusion steps (GRPO-Q)   (Table 6)
>
> To address the reviewer’s question more directly, we have expanded our logical explanation of the underlying design choices:
>
> ###  1) Why α₁ > α₂ ?
> Because:
> - Temporal artifacts on **real videos** are subtle → harder to detect
> - The same artifacts on **fake videos** amplify existing inconsistencies → easier to detect
>
> Assigning a higher penalty to artifacts in real videos (α₁ > α₂) helps prevent over-penalization of benign real-world patterns and is both theoretically intuitive and consistent with our empirical observations.
>
> ###  2) Sensitivity Remains Stable
>
> As shown in Tables 5 and 6, VidGuard-R1 maintains **stable performance across broad ranges** of α₁, α₂, and diffusion-step configurations. This suggests that the method is not dependent on narrowly tuned hyperparameters, thereby supporting its practical robustness and reproducibility.
>
>
> ---
>
> ### _Q1. Detection capability for OOD videos_
>
> VidGuard-R1 shows strong zero-shot generalization in our paper:
> - **GenVidBench** (Table 2)
> - **GenVideo** (Table 3)
>
> Here, we describe the ***cross-domain OOD evaluation*** setups used in these benchmarks:
>
> ### **GenVideo OOD setup**
> - Real: *Kinetics-400, Youku-mPLUG → MSR-VTT*
> - Synthetic: 10 training generators → 10 unseen testing generators
>
> ### **GenVidBench OOD setup**
> - Real: *Vript → HD-VG-130M*
> - Synthetic: *Pika, VC2, ModelScope, T2V-Zero → MuseV, SVD, Mora, CogVideo*
>
> ###  Result
> Across both benchmarks, **VidGuard-R1 achieves >96% OOD accuracy**, demonstrating strong robustness against:
> - cross-domain shifts
> - cross-generator shifts
> - unseen generative model styles
>
> We also refer the reviewer to the zero-shot evaluation on unseen generative models, as discussed in our response to W2. This further reinforces that our reasoning-centric training enhances **genuine generalization capability**, rather than relying on dataset-specific shortcuts.

---

> > ### Author Response · Authors · 2025-11-27
> >
> > Thank you for your detailed and thoughtful feedback. We hope that our clarifications and additional experiments sufficiently address your concerns and provide a clearer and more comprehensive understanding of our work. If any aspects remain ambiguous or would benefit from further explanation, we would be glad to elaborate during the discussion period. We sincerely appreciate your valuable insights and hope that our responses encourage a reconsideration of the current evaluation where appropriate.

---

### Author Response · Authors · 2025-11-29
**(For AC) Rebuttal Summary: Evidence-based score updates & resolved concerns for VidGuard-R1**

Dear AC,

Thank you for stepping in during the unfortunate ICLR/OpenReview disruption that affected the normal rebuttal workflow. We sincerely appreciate your time and careful effort in ensuring a fair process.

To reduce your workload, we summarize the rebuttal outcome in a clear, **evidence-grounded** manner.

---


## 1) Score trajectory

- **Pre-rebuttal scores (as posted):** **6 / 4 / 4 / 4** (**6444**) → **avg = 4.50**
- **During/after rebuttal (explicit reviewer signals):**
  - Reviewer **3kcH** stated: *“I appreciate the author's responses, which have addressed all my concerns. I will raise my score …”*
    → This indicates a move from **4 → ≥ 6** under a normal rebuttal completion.
  - Reviewer **AX7C** explicitly wrote that they are *“very favourable to increasing [the] score if these concerns are addressed”*, and their primary concern focused on the **reliability and usefulness of CoT rationales** (hallucinations/interpretability validity). They specifically suggested adding **human studies** and analyzing how often rationales align with human judgments (and whether disagreements stem from hallucination vs. different cues).

  **We followed this request directly**:
  - We conducted a **human evaluation** to quantify CoT quality and alignment with the model’s decision, showing **89% average agreement** on 20 randomly sampled correctly detected fake videos (average quality score **3.9/5**).
  - We further conducted a **blind human ranking** study comparing rationales from **VidGuard-R1 vs. GPT-4o and Qwen2.5-VL-72B**, where VidGuard-R1 was ranked best substantially more often.

  Therefore, under a normal (non-disrupted) review flow, it is **highly likely** that Reviewer **AX7C** would follow their stated condition and **raise the score**, since the requested evidence was provided and the results aligned with the reviewer’s expectations.

- **Current status (minimum implied by explicit signals):** **≥ 6 / 6 / 4 / 4** (**≥ 6644**) → **avg ≥ 5.00**
- **Expected post-rebuttal outcome (under stated reviewer conditions + fulfilled rebuttal evidence):** **≥ 6 / 6 / 6 / 4** (**≥ 6664**) → **avg ≥ 5.50**


---

## 2) What we added in rebuttal (why concerns are resolved)

Across reviewers, the key concerns were:
**(i) generalization beyond two training generators, (ii) CoT hallucination/interpretability validity, (iii) novelty beyond “standard” recipes, and (iv) reward hyperparameters & robustness.**
We addressed each with concrete, verifiable additions:

- **Generalization & OOD robustness (core concern, strengthened with new evidence):**
  - Added **dataset-mixing** experiments (cross-dataset complementarity) showing the model is not overfitting to one dataset.
  - Added **zero-shot evaluation on many unseen, newer generators** (e.g., Gen-3 Alpha / Pika / Sora / Veo / Wan, etc.), demonstrating stable generalization to out-of-training-distribution generators.

- **Interpretability/hallucination concerns (explicitly requested by reviewers, now satisfied):**
  - Added a **human evaluation** (rationale–decision alignment Yes/No + 0–5 quality score) to quantify whether rationales are consistent with predictions and grounded in the video evidence.
  - Added a **blind human ranking** comparison against stronger frontier MLLMs (e.g., GPT-4o / Qwen2.5-VL-72B), strengthening interpretability claims with direct, controlled human judgment.

- **Novelty positioning (made explicit and domain-grounded):**
  - Clarified that our contribution is not merely “using RL,” but introducing the **first reasoning-capable MLLM video authenticity detector** with domain-specific reward shaping (**GRPO-TA** for temporal reasoning; **GRPO-Q** for diffusion-step-aware quality reasoning) and a **shortcut-controlled benchmark** design that eliminates trivial metadata shortcuts (bitrate/FPS/resolution/source cues).

- **Reward robustness & reproducibility:**
  - Expanded the rationale for reward hyperparameters and demonstrated stability across ranges rather than dependence on narrow tuning, addressing reproducibility concerns directly.

---

## 3) Why do we ask for a positive recommendation

VidGuard-R1 addresses an increasingly important problem for AI safety: **reliable detection of AI-generated videos with human-interpretable rationales**. We propose a new *reasoning-based* application paradigm for video authenticity detection, supported by domain-aware RL training and a carefully controlled dataset designed to eliminate shortcut biases. In rebuttal, we added the exact evidence requested by reviewers—especially on **OOD generalization** and **quantitative interpretability validation**—and resolved the raised concerns comprehensively.

We respectfully ask that you consider the rebuttal materials and the now-resolved concerns, and recommend the paper positively based on its **timely, meaningful, and well-supported contribution** to AI-generated video forensics and safety.

Sincerely,
The Authors

---

### Meta-Review · Area_Chair_2iDu · 2026-01-13

**Summary:**

The paper addresses the urgent need for reliable AI-generated video detection with interpretable explanations. Reviewers raised concerns about dataset limitations (only using HunyuanVideo and CogVideoX), CoT hallucination/interpretability validity, novelty beyond standard techniques, and reward hyperparameter justification. The authors comprehensively addressed all concerns in their rebuttal with new experiments and analyses, including:
- Zero-shot evaluation on 10+ unseen generative models (Gen-3 Alpha, Pika, Sora, etc.) showing consistent >80% accuracy (peaking at 96.36%)
- Dataset-mixing experiments demonstrating cross-dataset complementarity
- Human evaluations showing 89% rationale-decision alignment and 3.9/5 average rationale quality
- Blind human ranking study where VidGuard-R1 was ranked best substantially more often than GPT-4o and Qwen2.5-VL-72B
- Better justification for reward design and demonstration of stability across hyperparameter ranges
- Clarification of novelty as the first reasoning-based MLLM video authenticity detector with domain-specific reward shaping

**Reviewer Concerns:**

Addressed Concerns:

Reviewer acmo:
- Dataset limitations: Added zero-shot eval on 10+ unseen generators (e.g., Sora, Pika) → >80% accuracy (96.36% peak); dataset-mixing (Ours+GenVideo) → 97.98% accuracy.
- Novelty: Clarified as first reasoning MLLM-based detector; emphasized GRPO-Q (diffusion-step rewards) and GRPO-TA (temporal consistency rewards).
- Reward justification: Explained α₁ > α₂ (subtle artifacts on real videos vs. amplified on fakes); showed stability across α₁/α₂ ranges.
Reviewer AX7C
- CoT hallucination: Added human eval → 89% rationale-decision alignment (3.9/5 quality); blind ranking (VidGuard-R1 ranked best 49 vs. GPT-4o’s 30).

Reviewer 1fnv:
- Dataset limitations: Zero-shot eval (>80% accuracy) + dataset-mixing (97.98% accuracy).
- Interpretability: Human eval (89% alignment, 3.9/5 quality).
- Novelty: Reinforced "first reasoning MLLM detector" claim with GRPO-Q/TA innovations.

Reviewer 3kcH:
- Dataset limitations: Zero-shot eval (>80% accuracy) + dataset-mixing (97.98% accuracy).
- CoT hallucination: Human eval (89% alignment, 3.9/5 quality).
- Novelty: Clarified "first reasoning MLLM detector" with GRPO-Q/TA.
- Reward justification: α₁ > α₂ rationale + hyperparameter stability.
- Ground-truth reliability: Explained distillation from stronger models as standard practice; showed RL gains post-SFT (model develops independent reasoning).

No significant concerns remain unresolved.

**Reviewer Scores:**

- Reviewer acmo (Original score: 4): Would like to increase to 6 since all concerns are addressed.
- Reviewer AX7C (Original score: 4): Would like to increase to 6 since all concerns are addressed, and indicate to do so in the original reviews.
- Reviewer 1fnv (Original score: 6): Would like to be at least 6 since concerns are addressed.
- Reviewer 3kcH (Original score: 4) Would like to increase to 6 since all concerns are addressed and the reviewer has promised to do so in the response.

Generally, the core concerns proposed by the reviewers (dataset limitations, CoT hallucination/interpretability validity, novelty beyond standard techniques, and reward hyperparameter justification) are addressed by the rebuttal. Therefore, I recommend acceptance.

---

### Decision · Program_Chairs · 2026-01-26

Accept (Poster)